

# Sea ice concentration impacts dissolved organic gases in the Canadian Arctic

Charel Wohl[1,2,3,*], Anna E. Jones[3], William T. Sturges[2], Philip D. Nightingale[1,2,4], Brent Else[5], Brian J. Butterworth[6, 7], Mingxi Yang[1]

[1]Plymouth Marine Laboratory, Plymouth, PL1 3DH, United Kingdom
[2]School of Environmental Sciences, University of East Anglia, Norwich, NR4 7TJ, United Kingdom
[3]British Antarctic Survey, Cambridge, High Cross, Madingley Road, CB3 0ET, United Kingdom
[4]Sustainable Agriculture Systems, Rothamsted Research, North Wyke, Devon, EX20 2SB, United Kingdom
[5]University of Calgary, Calgary, Alberta, T2N 1N4, Canada
[6] Cooperative Institute for Research in Environmental Sciences, University of Colorado, Boulder, Colorado, USA
[7] NOAA Physical Sciences Laboratory, Boulder, Colorado, USA
[*]Now at : Institut de Ciències del Mar, Barcelona, 08003, Spain

*Correspondence to*: Charel Wohl (chwo@pml.ac.uk) and Mingxi Yang (miya@pml.ac.uk)

**Abstract.** The marginal sea ice zone has been identified as a source of different climate active gases to the atmosphere due to its unique biogeochemistry. However, it remains highly undersampled and the impact of changes in sea ice concentration on the distributions of these gases is poorly understood. To address this, we present measurements of dissolved methanol, acetone, acetaldehyde, dimethyl sulfide and isoprene in the sea ice zone of the Canadian Arctic from the surface down to 60 m. The measurements were made using a Segmented Flow Coil Equilibrator coupled to a Proton Transfer Reaction Mass Spectrometer. These gases varied in concentrations with depth, with the highest concentrations generally observed near the surface. Underway (3-4 m) measurements showed broadly higher concentrations in partial sea ice cover compared to ice-free waters. The large number of depth profiles at different sea ice coverages enables proposition of the likely dominant production processes of these compounds in this area. Methanol concentrations appear to be controlled by specific biological consumption processes. Acetone and acetaldehyde concentrations are influenced by the penetration depth of light and the mixed layer depth, implying dominant photochemical sources in this area. Dimethyl sulfide and isoprene both display higher surface concentrations in partial sea ice coverage compared to ice-free waters due to ice edge blooms. Dimethyl sulfide concentrations sometimes display a subsurface maximum in ice -free conditions, while isoprene displays more reliably a subsurface maximum. Surface gas concentrations were used to estimate their air – sea fluxes. Despite obvious in situ production, we estimate that the sea ice zone is absorbing methanol and acetone from the atmosphere. In contrast, DMS and isoprene are consistently emitted from the ocean, with marked episodes of high emissions during ice-free conditions, suggesting that these gases are produced in ice-covered areas and emitted once the ice has melted. Our measurements show that the seawater concentrations and air-sea fluxes of these gases are clearly impacted by sea ice concentration. These novel measurements and insights will allow us to better constrain the cycling of these gases in the polar regions and their effect on the oxidative capacity and aerosol budget in the Arctic atmosphere.



## 1 Introduction

The Arctic is an important part of the global climate system and is warming faster than the rest of the world (Dai et al., 2019).
One of the most obvious signs of the warming/changing Arctic is the changes in sea ice. Sea ice is rapidly decreasing in extent
and concentration (Meier et al., 2014; Wang et al., 2020b), melting earlier, and freezing up later in the season (Markus et al.,
2009; Wang et al., 2020b). The effects of these changes on the marine biogeochemistry and trace gas emissions are poorly
known (Huntington et al., 2019), largely due to a lack of measurements. Understanding these effects is particularly relevant as
areas of open water and open pack ice, which are becoming more frequent features of the Arctic, have been associated with
new particle formation (Collins et al., 2017; Dal Íosto et al., 2018). This suggests a possible negative feedback-link between
changes in sea ice and Arctic climate (Dall' Osto et al., 2017), potentially leading to reduced Arctic warming (Mahmood et
al., 2018; Paasonen et al., 2013). Such a link could proceed via emission of a cocktail of trace gases from the ice uncovered
waters, leading to increased cloud condensation nuclei (Collins et al., 2017; Köllner et al., 2017; Mungall et al., 2017). Most
of the new particles formed in the Canadian Arctic (Tremblay et al., 2019) and the remote ocean (Zheng et al., 2020) appear
to consist of organic material and sulfate (oxidation product of dimethyl sulfide).

In this paper, we focus on the impact of sea ice concentration on the seawater concentrations and air – sea fluxes of methanol,
acetone, acetaldehyde, dimethyl sulfide (DMS) and isoprene. Global ocean fluxes of methanol, acetone, acetaldehyde and
isoprene are highly uncertain (Arnold et al., 2009; Bates et al., 2021; Wang et al., 2019, 2020a). Therefore these measurements
will help to constrain the ocean emissions of these gases not only in the polar region, but also globally.

All of these gases have known sources in seawater, which are potentially enhanced in the marginal sea ice zone. Methanol is
produced by phytoplankton (Kameyama et al., 2011; Mincer and Aicher, 2016) and consumed by microbes (Dixon et al., 2011,
2013; Sargeant et al., 2016). Acetone is thought to be produced primarily from photochemical reactions (De Bruyn et al.,
2011a; Dixon et al., 2013; Kieber et al., 1990) and consumed by microbes (Dixon et al., 2014a). A biological source of acetone
has also been suggested from culture experiments (Davie-Martin et al., 2020; Halsey et al., 2017) and correlations of field data
(Schlundt et al., 2017), though it is thought to be small. Acetaldehyde is produced by photochemistry (Dixon et al., 2013; Zhu
and Kieber 2018; Kieber et al., 1990; De Bruyn et al., 2011a) and in a light dependent fashion from phytoplankton (Davie-
Martin et al., 2020; Halsey et al., 2017). Acetaldehyde is consumed very rapidly by microbes, giving it a lifetime of a few
hours in seawater (de Bruyn et al., 2017, 2021; Dixon et al., 2014b). Acetone and methanol have highly variable lifetimes in
the ocean mixed layer (5-55 days for acetone and 10-26 days for methanol (Dixon et al., 2013)), with shorter lifetimes generally
observed in coastal areas (de Bruyn et al., 2013; Dixon et al., 2014a). Production of DMS in seawater is complex and has been
the subject of many prior studies (Simó, 2001; Zhang et al., 2019). Different phytoplankton produce dimethyl
sulfoniopropionate (DMSP, a key precursor for DMS) and DMS at widely different rates (Sheehan and Petrou, 2020). The
largest sink of DMS in seawater is biological consumption by bacteria (Kiene and Bates, 1990; Yang et al., 2013b), giving it
a turnover time between 0.5 and 2 days. Isoprene is thought to be produced by a large range of phytoplankton in the ocean
(Hackenberg et al., 2017; Shaw et al., 2010a) and is often parametrised with sea surface temperature and surface chlorophyll





*a* (Chl *a*) concentration (Hackenberg et al., 2017; Ooki et al., 2015; Rodriguez-Ros et al., 2020). The largest sink of isoprene to the water column is thought to be air – sea exchange (Booge et al., 2018; Palmer and Shaw, 2005), giving it a lifetime of around 10 days (Booge et al., 2018).

Sea ice may directly or indirectly affect the production and consumption of these gases in seawater. In general, the water column in the high Arctic displays very shallow light penetration depths (Pavlov et al., 2015), largely due to the input of light absorbing molecules from riverine input (Granskog et al., 2015). Seasonal sea ice meltwater input leads to deeper light penetration depths (Granskog et al., 2015). This affects the biological and photochemical production of many of the compounds discussed here. Sea ice its self is generally poorly transmissible to light, but breaks in the ice cover and melt ponds allow more light to pass, creating a very heterogenous light field (Massicotte et al., 2018). Sea ice also plays an important (but poorly understood) role in gas exchange (e.g. Butterworth and Miller (2016); Loose et al. (2014)) which in turn would be expected to impact the source/sink terms for many of these compounds.

The Arctic Ocean and the sea ice zone represent particularly undersampled regions with no existing measurements of seawater concentrations of methanol, acetone and acetaldehyde in partial sea ice cover. Based on atmospheric measurements and correlations with DMS, the Canadian Arctic sea ice zone has been suspected to be a sink for methanol and acetone (Sjostedt et al., 2012), and a source for other oxygenated VOCs produced in the sea surface microlayer from photochemical activity (Mungall et al., 2017, 2018). Atmospheric measurements in east Greenland find that the correlation between DMS and acetone air mixing ratios changes depending on the season (Pernov et al., 2020), pointing towards seasonal changes in seawater biogeochemistry affecting emissions of these gases.

In this paper, we present depth profile (0-60 m) and underway (~5 m) seawater measurements of methanol, acetone, acetaldehyde, DMS and isoprene in the Canadian Arctic during boreal summer (July-August 2017). Importantly, these data enable assessment of the air – sea fluxes of these gases. Combining these measurements, we investigate the impact of sea ice concentration on the dissolved concentrations of these VOCs.

## 2 Experimental

### 2.1 Description of the cruise sampling

Depth profile and underway measurements of dissolved gas concentrations in the sea ice zone of the Canadian Arctic were measured on board the ice breaker CCGS *Amundsen*. The measurements were taken between 17/07/2017 and 08/08/2017 (Cruise 1702, leg 2b) (Figure 1).



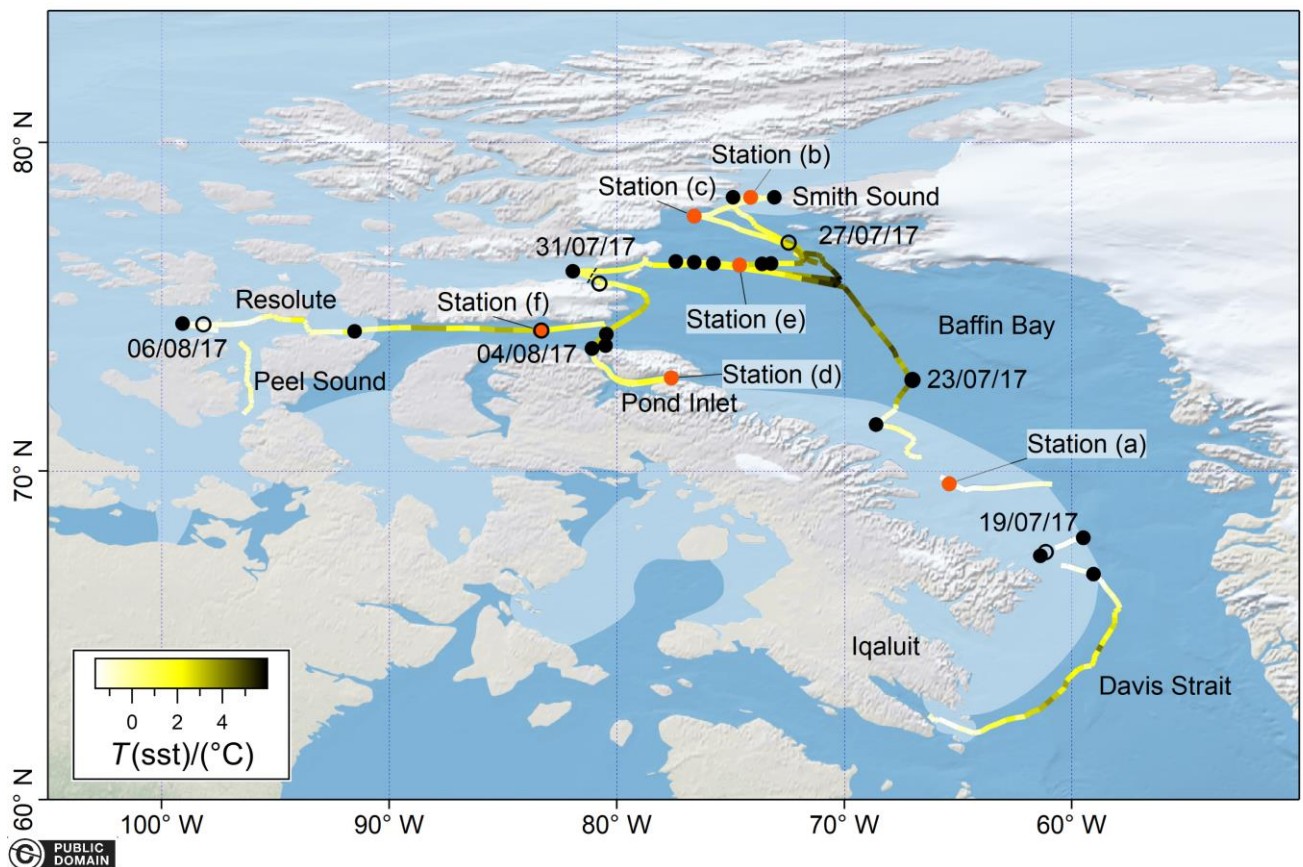

**Figure 1 Cruise track of the sampling undertaken in the Arctic sea ice zone coloured by sea surface temperature (sst). Sampling dates are indicated as hollow circles marked with the date. The location of each CTD station where sampling was undertaken is indicated as a black closed dot. Highlighted CTD stations are indicated as orange closed dots and labelled by station number. Interruptions in the cruise track and underway auxiliary data are due to failures in the ship underway logging system (Amundsen Science Data Collection, 2017). All the map data were created from public domain GIS data found on the Natural Earth website (http://www.naturalearthdata.com, last access: 15 April 2021). They were read into Igor using the Igor GIS XOP beta. The sea ice covered area is approximately indicated as a shaded area due to difficulties of conveying this information for a month-long deployment. Details on the sea ice concentration data used for analysis can be found in the text (Sect. 2.1).**

The research vessel travelled from Iqaluit northwards through Davis Strait and Baffin Bay to reach Smith Sound. In this area, more intense depth profile sampling was carried out. The vessel then travelled to Pond Inlet and Resolute. Sampling ended south of Resolute in Peel Sound.

Depth profiles of dissolved gas concentrations were measured from the Rosette-mounted Niskin bottles as discrete samples from the near surface (about 2 m) to 60 m depth at a total of 21 stations as per availability. A range of sensors were mounted on the Rosette frame to measure biogeochemical variables from 2 m downwards, such as oxygen concentration (monitored using a Seabird 43), conductivity (Seabird 4), Chl $a$ (Sea point Chlorophyll Fluorometer), PAR irradiance (QCP-2300 Biospherical), temperature (Seabird 3 plus) and pressure (Paroscientific Digiquartz).



When logistically feasible at station, a handheld vertical 5 dm$^3$ Niskin bottle was deployed off the front starboard side of the ship to sample approximately the top 30 cm from the ocean surface. This was done by bringing the Niskin bottle up from approximately 3 m and firing it just before it reached the surface using rope marks as depth indicators, similar to described in Ahmed et al. (2020). Samples collected with this method are marked in the data presentation. This sampling was preceded by the deployment of a handheld CTD logger (RBR XR-420) to characterise the salinity and temperature in the upper meters of the ocean. Data from this logger is presented at 0.5 m resolution for the 2 m near the surface.

A range of biogeochemical parameters were monitored continuously underway, including sea surface temperature (sst) (Sea Bird SBE 38), sea surface salinity (sss) (Sea Bird SBE 45 MicroTSG Thermosalinograph) and Chl *a* fluorescence (Wetlabs WETStar Fluorometer). When not used for discrete sampling, the VOC measurement system (see next section) was also used for continuous sampling of the underway water.

Wind speed was measured from a meteorological tower (approximately 16 m) located on the foredeck of the ship, similar to that described in Ahmed et al. (2019) and corrected to 10-m neutral wind speed by accounting for speed of ship passage.

The AMSR2 passive microwave sea ice concentration (SIC) satellite product (daily, 3.125 km x 3,125 km resolution) (Spreen et al., 2008) was used to create a time series of sea ice concentration along the cruise track. This product is chosen due to its high spatial and temporal resolution as well as for complete coverage of the cruise track. From each daily satellite image, the sea ice concentration of the grid cell where the ship was located during that hour was extracted. The sea ice cruise track time series deduced from the AMSR2 satellite product and visual ship-based ice fraction observations largely agree and show no major systematic bias (Supplement S1). Due to its wider spatial coverage, the satellite timeseries is used to infer the impact of seasonal sea ice melt on the underway dissolved concentrations of VOCs. Visual SIC observations were made during CTD casts directly from the CTD launch point on deck and thus those estimates were used to interpret vertical profile measurements. In the analysis below, we mainly assess how the depth profiles and underway seawater VOC concentrations vary with SIC. From this, the impact of seasonal sea ice melt on VOC cycling is inferred.

## 2.2 Dissolved gas measurements

A Segmented Flow Coil Equilibrator (SFCE) coupled to Proton Transfer Reaction-Mass Spectrometer (PTR-MS) was used to measure dissolved gases in seawater (Wohl et al., 2019). The SFCE-PTR-MS system was set up in one of the labs located near the front of the ship with access to an underway water tap from the ship's main underway water supply (located at 3-4 m depth). Discrete water samples (for depth profiles) were taken from the Niskin bottles in gas-sensitive fashion into 900 cm$^3$ glass bottles with glass caps. For discrete VOC measurements, the SFCE sampled from the bottom of these glass bottles. At a water flow rate of about 100 cm$^3$ min$^{-1}$, this water volume was enough for a stable measurement (i.e. average of ~ 6 minutes), where the top 5 cm of water in the glass bottle was not sampled due to possible atmospheric contamination. For underway measurements, the SFCE sampled from the bottom of a glass bottle in the sink, which was rapidly overflowed with the ship's underway water. During periods of high sea ice cover, as per decision by the ship's crew, the underway water inlet was turned



off. The underway water flow rate was continuously monitored by the ship's crew and used for data quality control. For more information on the installation on board and comparison of CTD to underway measurements, please refer to Wohl et al. (2019). The computation of dissolved gas concentration specific to this deployment is laid out in the Supplement S2. Comparisons between near-surface CTD and underway measurements suggested an initial acetaldehyde contamination in the CTD rosette bottles due to the use of an air duster aerosol spray used near the rosette. The other VOCs were not affected. After use of the

spray was stopped on 26/07/2017, the acetaldehyde contamination in the CTD measurements immediately disappeared. Thus, acetaldehyde CTD measurements collected prior to 26/07/2017 are not included in this analysis.

## 3 Results

### 3.1 Depth profile distributions

To illustrate the effect of SIC on the depth profile distributions of these dissolved gases, we first focus on the shape of their

depth profiles. A discussion of the effect of sea ice on the absolute dissolved gas concentrations follows, which predominantly considers the underway measurements leading to a much greater sample size. Figures 2 to 6 are overview plots , which focus on the VOC depth profile shapes along with corresponding auxiliary data. The casts have been grouped in panels by SIC (indicated at the bottom of the panel) and staggered along the x-axis by sampling order for ease of viewing. The sequence of the casts is the same for all compounds except acetaldehyde, (due to missing depth profile data). Scale bar for VOC

concentrations and auxiliary data are also included. Additionally, six casts are highlighted in a separate figure along with more detailed auxiliary data to show absolute concentrations and allow comparisons between casts (Figure 7). These profiles are chosen from careful examinations of the overview plots as they; (i) represent the typical effect of sea ice on these compounds, (ii) present higher vertical resolution near the surface and (iii) contain acetaldehyde concentrations which could not be determined for all profiles. The highlighted casts are also marked in the overview plots. Seawater samples collected with a

handheld Niskin from about 30 cm depth are marked as a blue diamond. Density, temperature and salinity data for the surface 2m are only presented when near surface sampling was carried out.



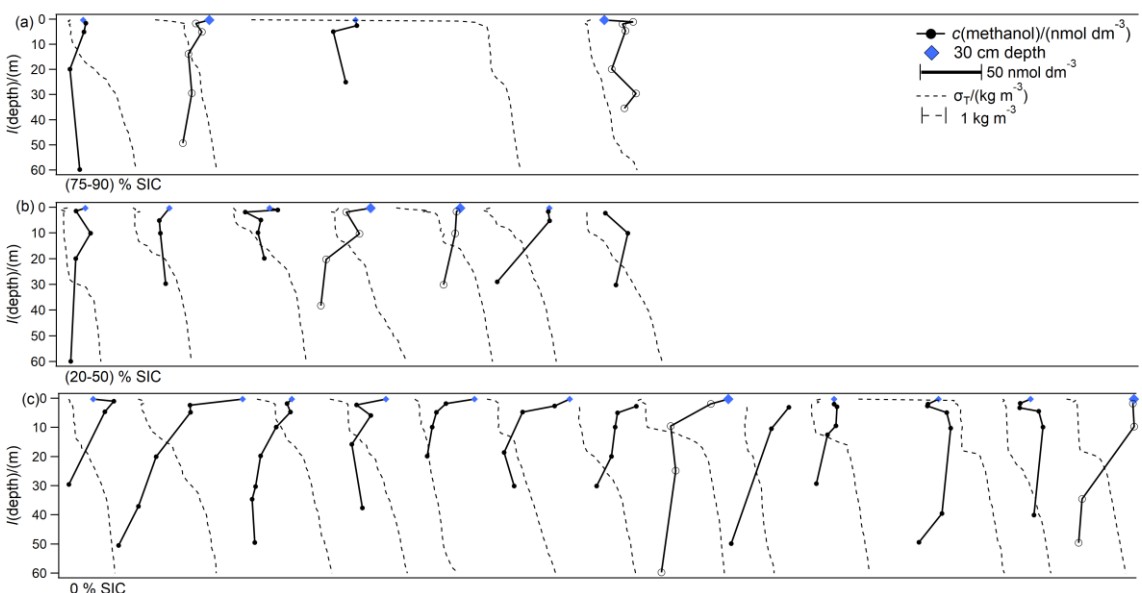

**Figure 2 Overview plot displaying the shape of all methanol and density ($\sigma_T$) depth profiles, grouped by SIC and staggered along the x-axis for ease of viewing. Panel labels indicate the SIC bin. The scale bars for methanol and density in panel (a) apply also to panels (b) and (c). Profiles with hollow markers are highlighted in Figure 7.**

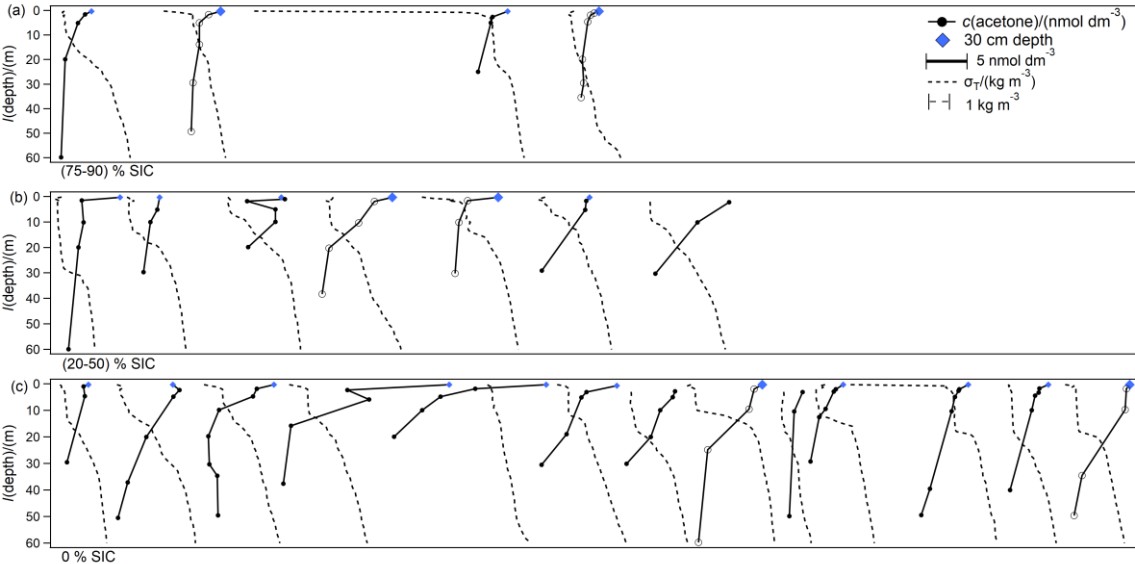

**Figure 3 Overview plot displaying the shape of all acetone and density ($\sigma_T$) depth profiles, grouped by SIC and staggered along the x-axis for ease of viewing. Panel labels indicate the SIC bin. The scale bars for acetone and density in panel (a) apply also to panels**





**(b) and (c). Profiles with hollow markers are highlighted in Figure 7.**

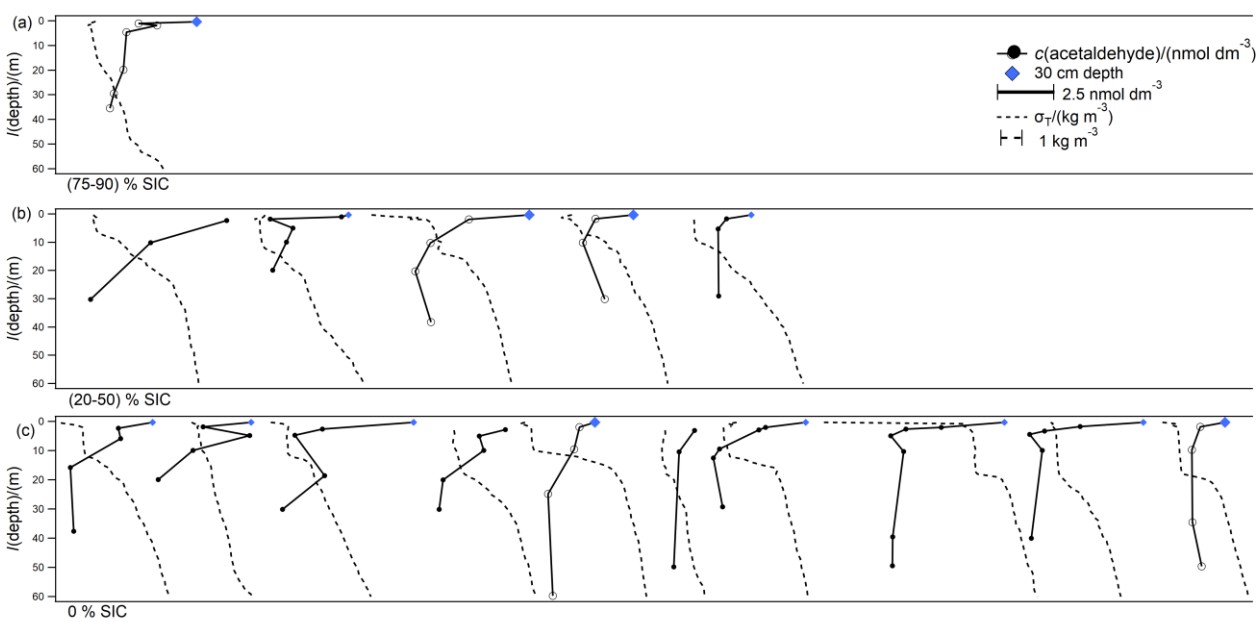


**Figure 4 Overview plot displaying the shape of all acetaldehyde and density ($\sigma_T$) depth profiles, grouped by SIC and staggered along the x-axis for ease of viewing. Panel labels indicate the SIC bin. The scale bars for acetaldehyde and density in panel (a) apply also to panels (b) and (c). Profiles with hollow markers are highlighted in Figure 7.**

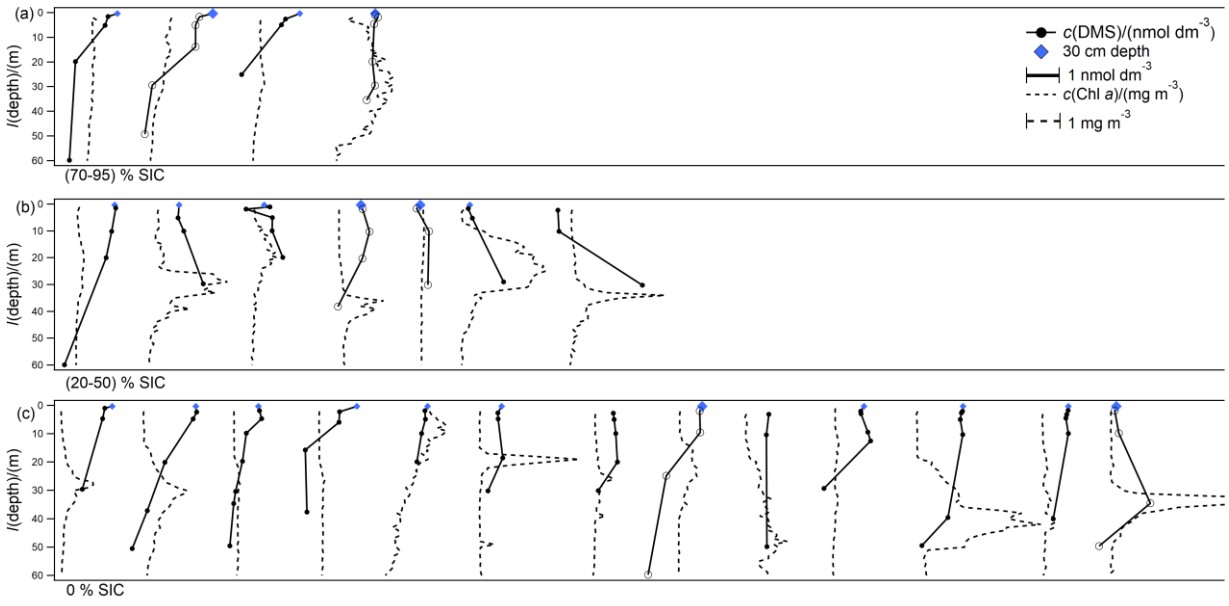

**Figure 5 Overview plot displaying the shape of all DMS and Chl _a_ depth profiles, grouped by SIC and staggered along the x-axis for ease of viewing. Panel labels indicate the SIC bin. The scale bars for DMS and Chl _a_ in panel (a) apply also to panels (b) and (c). Profiles with hollow markers are highlighted in Figure 7. One of the Chl _a_ profiles is cut off in panel (c) for scale purposes.**





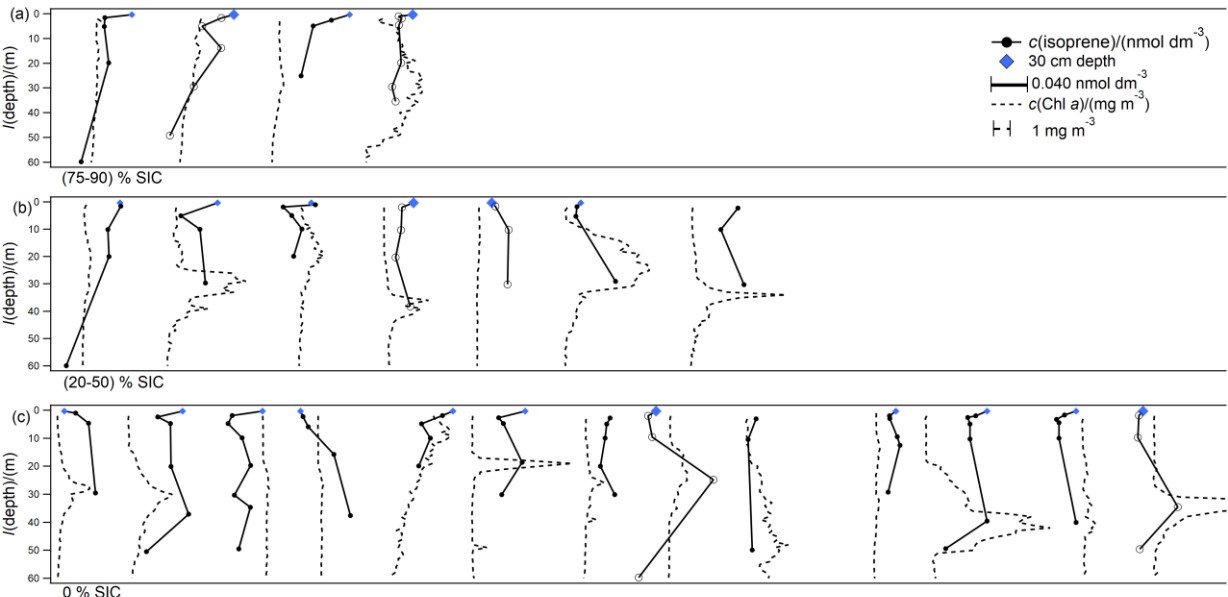

**Figure 6 Overview plot displaying the shape of all isoprene and Chl *a* depth profiles, grouped by SIC and staggered along the x-**
**axis for ease of viewing. Panel labels indicate the SIC bin. The scale bars for isoprene and Chl *a* in panel (a) apply also to panels**



**(b) and (c). Profiles with hollow markers are highlighted in Figure 7. One of the Chl *a* profiles is cut off in panel (c) for scale**

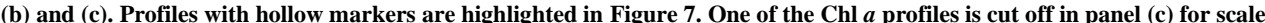

**purposes.**

**Figure 7** Depth profile concentrations arranged by decreasing SIC. Geographical locations of the stations a-f are indicated in Figure 1.. SIC at the time of sampling were (a) 90 %, (b) 75 %, (c) 50 %, (d) 20 %, (e) 0 %, and (f) 0 %, where 0 % indicates ice free at the time of sampling. (Aetal.=Acetaldehyde, Temp.=Temperature, * Chl a increases up to 13 mg dm$^{-3}$ at the Chl a max). Error bars show measurement noise. (Amundsen Science Data Collection, 2017). Limited measurements of acetaldehyde during the early part of the cruise are due to contamination from the CTD and the single measurement came from the handheld Niskin bottle (30 cm depth).





Here we briefly discuss the effect of sea ice concentration on water column structure and biogeochemistry to provide context

for the dissolved gas measurements. As shown in Figure 7 and for example Fig. 2, the stations with near full ice cover/during ice break up (75 to 90 % SIC, following the definition of ice breakup by Ahmed et al. (2019)) show fairly small density and salinity gradients between 2 and 60 m depth, perhaps due to limited wind-driven mixing. In these casts, high concentrations of Chl *a* were found at 2 m and Chl *a* concentrations then gradually decreased down to 60 m (Fig. 5 and Fig. 7).

Stations with lower ice coverage (20 to 50 % SIC) tend to display a more defined, very shallow mixed layer of similar density

and salinity between 2 and ca. 10 m. A deep Chl *a* maximum is observed just below the density gradient (at ca. 10 m depth, Figure 7), similar to previous observations (Martin et al., 2010). This deep Chl *a* maximum at some of the stations (Figure 7) is characterised by very high levels of biological activity (Ardyna et al., 2013; Barber et al., 2015; Burgers et al., 2020).

Ice-free casts (0 % SIC) display a deeper (from 2 to ca. 20 m depth) and warmer mixed layer of similar density and salinity (Figure 7) – useful indicators for how long these stations have been ice-free (Shadwick et al., 2013). The density gradient at

the base of this mixed layer is much larger at these ice-free stations, and many of the profiles display a very pronounced deep Chl *a* maximum just below the mixed layer, while surface Chl *a* is generally lower (Figure 7). Even ice-free areas display very low salinities (between 32 and 28, Figure 7), which is typical for this region (McLaughlin et al., 2004) and indicates that that the waters sampled here are heavily influenced by sea ice melt and riverine discharge.

Salinity and density measurements between the surface and 2 m show that about half of the casts display lower density waters

in the top 1 m (e.g. Fig. 2) coinciding with lower salinity and sometimes higher temperature (Fig. 7). In the data shown here, this surface stratified layer tended to be more common in casts with partial sea ice cover, but also occurred when no sea ice was present at the time of sampling (Figure 7 and e.g. Fig. 2). We speculate that this surface stratified layer is largely due to sea ice melt in this region (Ahmed et al., 2020; Burt et al., 2016; Miller et al., 2019).

We next present the underway measurements of these gases. Combining the depth profiles with the underway measurements,

we discuss how the seawater concentrations of the trace gases change with different SIC in Section 4.

## 3.2 Underway measurements

Compared to CTD measurements at stations, the underway measurements presented in this section have a much higher temporal and spatial coverage. Hence, they can be used to derive more robust statistics and investigate the effect of sea ice on absolute concentrations in surface seawater. These underway measurements are compared to previous observations in other

parts of the ocean and also used to derive correlations with ancillary measurements.

Underway seawater concentrations of methanol, acetone, acetaldehyde and isoprene are presented in Figure 8 along with the concentrations measured from the 5 m Niskin bottle. Underway sst, SIC, Chl *a* and sss are also presented.





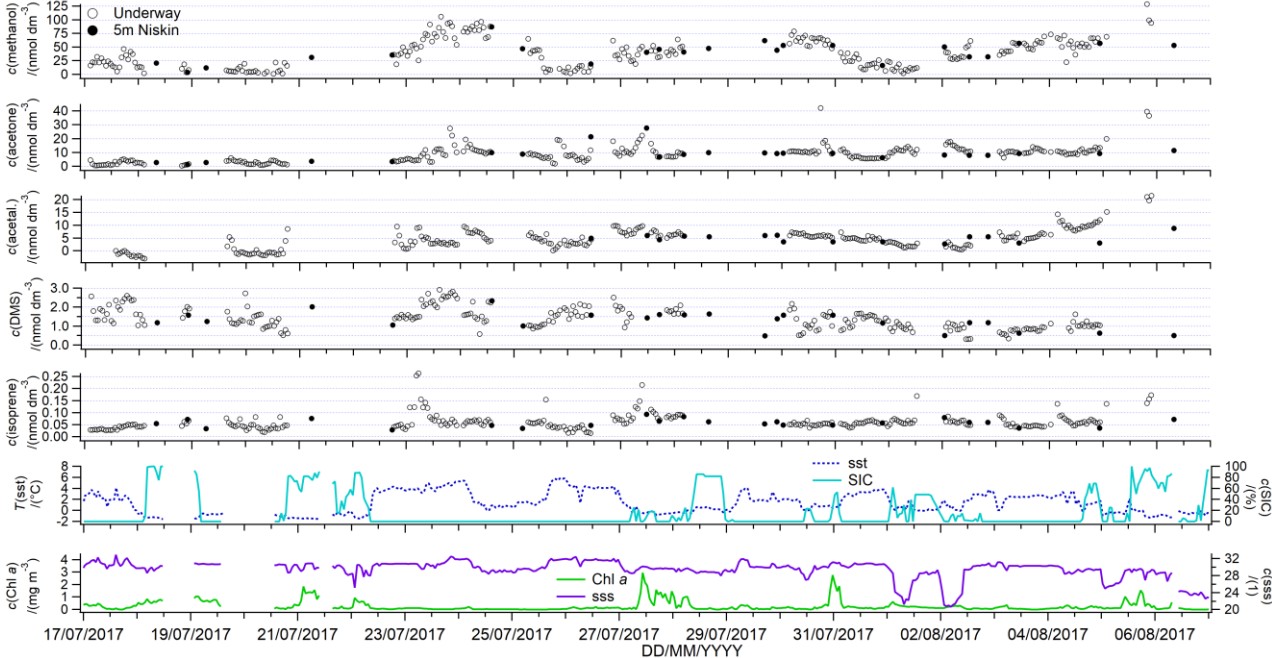

**Figure 8 Underway surface water concentrations of dissolved gases. Plotted along are underway SIC, SST, Chla and salinity**
**(Amundsen Science Data Collection, 2017).**

The underway timeseries (Figure 8) shows generally higher Chl *a* surface concentrations in partial ice cover compared to ice-free or full ice cover. These may be in part due to under ice phytoplankton blooms or ice breakup-blooms, which are frequent features of the Arctic sea ice zone (Barber et al., 2015; Levasseur, 2013; Perrette et al., 2011).

To further investigate the effect of sea ice on surface seawater concentrations of these compounds, the measured concentrations
are plotted against sea ice concentration at the time of sampling and bin averaged to 10 % SIC bins (Figure 9). A total of 61 hourly underway surface seawater measurements were taken in partial ice cover, which represents 23 % of the underway measurements shown here.





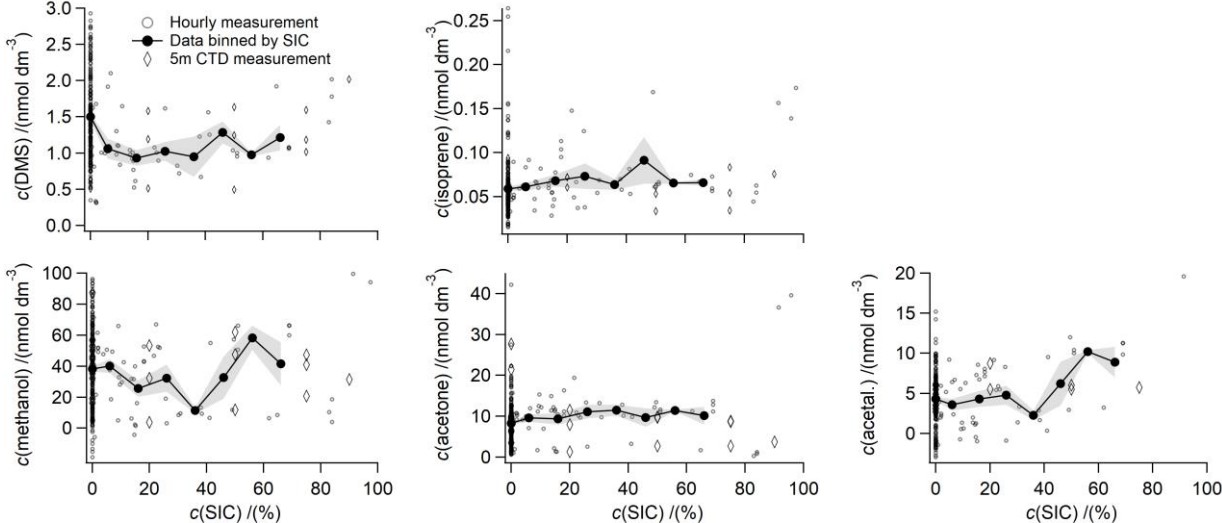

**Figure 9 Underway seawater concentrations plotted against SIC and binned to 10 % SIC bins. The standard error of the SIC bin is**
**indicated as grey shaded area. SIC bins have only been calculated for SIC up to 70 % due to scarcity of data at higher SIC.**

## 4 Discussion

### 4.1 Methanol

Casts with near full ice cover (75 to 90 % SIC) displayed somewhat similar concentrations of methanol throughout the top 60
m (Fig. 2a), while partially ice covered (20 to 50 % SIC) and ice-free casts displayed higher methanol concentrations in the
mixed layer and near the surface (Fig. 2b and Fig. 2c). Many of the ice-free casts that display lower density seawater near the
surface also tended to show higher methanol concentrations at 30 cm (Fig. 2c). The few methanol profiles collected in the
temperate and tropical Atlantic indicate generally higher concentrations of methanol within the mixed layer than below (Beale
et al., 2013; Williams et al., 2004; Yang et al., 2014b). Higher methanol concentrations near the surface could be due to air to
sea deposition of methanol in ice-free-conditions. In seawater, methanol is produced by a large range of phytoplankton (Davie-
Martin et al., 2020; Mincer and Aicher, 2016) and consumed by bacteria (Dixon and Nightingale, 2012; Sargeant et al., 2016).
Higher concentrations near the surface at stations of low ice coverage appear to be consistent with a biological source of
methanol in seawater as ice-free areas typically display higher net community production (Burgers et al., 2020) and most
primary productivity tends to occur within the 30 m near the surface (Arrigo et al., 2011). We observe no obvious relationship
between methanol concentration and Chl $a$ in this dataset. This might be because the balance between biological production
and consumption depends on the phytoplankton and bacteria species present (Mincer and Aicher, 2016; Sargeant et al., 2016).
Methanol concentrations near the surface tend to be quite variable, which could be because biological consumption rates are
also highly variable with depth (Dixon and Nightingale, 2012). Indeed, some ice-free casts display higher methanol
concentration in the top 2m and 30 cm compared to the rest of the mixed layer (Figure 2c, Figure 7).The shapes of the methanol



depth profiles are remarkably similar to other compounds that display photochemical sources (acetone, acetaldehyde), which
suggests a possible role of light in methanol production. Since the presence or absence of sea ice is a major control on the
amount of light in the upper ocean, our observations of higher methanol at lower sea ice concentration also suggests a light
dependent source. Previous experiments suggest that direct photochemical production of methanol is negligible (Dixon et al.,
2013). Higher light intensity has been shown to lead to higher biological methanol production rates (Halsey et al., 2017).
Though those experiments used visible light, which penetrates deeper into the water column ($\approx 40 - 50$ m (Massicotte et al.,
2018), Figure 7). The near surface enhancement in methanol we observed here suggests that this methanol concentration
gradient may be caused by ultra violet (UV) light which penetrates to around 2-7 m (Tedetti and Semperv, 2006). We speculate
that the very near surface enhancement in methanol concentrations (within the top $\approx 2$ m) could have been in part due to cell
lysis caused by damaging UV light. Cell lysis has been suspected to possibly interfere with previous methanol production rate
measurements (Davie-Martin et al., 2020). Lethal levels of UV light have been observed to depths of 2-3 m in the Arctic
(Tedetti and Semperv, 2006).

The mean underway surface seawater concentration of methanol was 38 nmol dm$^{-3}$ and the median was 36 nmol dm$^{-3}$. This is
within the range of previous seawater measurements (Beale et al., 2013; Kameyama et al., 2010; Williams et al., 2004) and
the mean is similar to measurements at UK shelf seas (Beale et al., 2015) and in the temperate Atlantic (Yang et al., 2013a).
Our mean concentration is about twice as high as previous measurements in the Labrador Sea in October (Yang et al., 2014a),
possibly due to higher seasonal biological activity during the cruise presented here (e.g. Davie-Martin et al. (2020)). Another
reason for higher methanol concentrations during this cruise could be

Methanol concentrations displayed a large range in concentrations (below limit of detection up to 129 nmol dm$^{-3}$) which were
better resolved relative to previous discrete measurements due to the use of high resolution underway sampling. This suggests
that methanol production and consumption processes are not always tightly coupled and instead governed by specific processes
in different locations. Instead of being controlled by wide spread sources, such as production by a large range of phytoplankton
(Mincer and Aicher, 2016), it is possible that methanol concentrations in the sea ice zone are heavily influenced by oxidation
rates. Methanol oxidation rates tend to be (a) highly variable (Dixon et al., 2011; Dixon and Nightingale, 2012) and (b)
influenced by the microbial species present (Sargeant et al., 2016, 2018). Methanol oxidation rates have also been shown to
(c) influence seawater methanol concentrations in coastal waters (Beale et al., 2015). Underway methanol concentrations do
not appear to vary with on SIC itself (Fig. 9). The presence or absence of sea ice at the time of sampling appears to influence
methanol concentrations more strongly.

## 4.2 Acetone

The stations with highest SIC (75 to 90 %) displayed highest concentrations at the surface, which decreased rapidly with depth
and reached a near-constant value at around 5 m (Fig. 3a). At stations with lower ice coverage (20 to 50 %) (Fig 3b), acetone
concentrations were also elevated at the surface and decreased with depth, reaching a near constant value at around 20-30 m.
At some ice-free stations with a well-defined mixed layer, the concentrations of acetone in the mixed layer were very





homogenous and higher than below the mixed layer (Figure 3c, Figure 7). Most of the casts that display stratification within the 2 m near the surface also display higher acetone concentrations at 30 cm compared to at 2 or 5 m. The acetone profiles could be explained by dominant photochemical production of acetone (De Bruyn et al., 2011b; Dixon et al., 2013) and the

penetration depth of light, which is influenced by sea ice concentration and freshwater input. UV light is required for the photochemical production of acetone and UV light is rapidly absorbed within the first 2-7 m of the Arctic water column (Tedetti and Semperv, 2006). Thus the photochemical production rate is likely higher in the first 2-7 m, leading to higher concentrations of acetone near the surface. The fine scale vertical gradients of acetone in the ice-covered stations and in the 2 m near the surface are probably preserved due small differences in density preventing mixing between the different depths. At

lower sea ice concentration, sea ice meltwater leads to stratification and dilutes light absorbing molecules, leading to deeper light penetration depths (Granskog et al., 2015). The surface stratified layer receives most of this radiation (Granskog et al., 2015), leading to small scale concentration gradients of acetone with highest concentrations at the surface. As a more defined mixed layer forms and the sea ice concentration decreases, UV light can penetrate even deeper into the water column and leads to production of acetone at deeper depths. In some of the ice-free casts, acetone is likely produced at the surface and mixed

deeper, forming a fairly homogeneous profile within the mixed layer. The ice-free casts with homogeneous acetone concentrations in the mixed layer are similar to previous measurements in the open ocean (Beale et al., 2013; Williams et al., 2004). We do not observe an obvious relationship between acetone and Chl $a$ in these depth profiles. If light dependent biological production of acetone were an important process in the sea ice zone, we would have expected to detect substantial acetone concentrations near depths of peak Chl $a$ and down to the penetration depth of visible light ($\approx 40 - 50$ m (Massicotte

et al., 2018), Figure 7)) required for biological activity. In Baffin Bay area, potentially up to 70 % of the primary productivity is occurring at the deep Chl $a$ maximum (Burgers et al., 2020). Earlier incubation experiments suggest that biological production of acetone is negligible (Dixon et al., 2013), while more recent field campaigns (Schlundt et al., 2017) and culture experiments (Halsey et al., 2017) suggest that acetone may have a considerable biological source. While it is possible that some of the acetone we observed below $\approx 10$ m is derived from biological activity, it appears that photochemistry is the

dominant acetone source here.

The mean (median) underway seawater acetone concentration measured during this deployment is 8.9 (9.1) nmol dm$^{-3}$ with a large range of 0.3 to 46.7 nmol dm$^{-3}$ (Figure 8). The mean concentration is similar to previous measurements in UK coastal waters (Beale et al., 2015) and to previous high latitude measurements in the Labrador Sea in October (Yang et al., 2014a) and the Fram Straight in June/July (Hudson et al., 2007). Concentrations from this deployment are generally lower than other

temperate and tropical open ocean measurements (Beale et al., 2013; Kameyama et al., 2010; Marandino et al., 2005; Schlundt et al., 2017; Williams et al., 2004; Yang et al., 2014b). Acetone surface seawater concentrations have been shown to vary seasonally at a temperate site (highest concentrations in summer (Beale et al., 2015), possibly due to greater photochemical production and slower consumption during the warmer months (Dixon et al., 2014b)). Using a machine learning technique, Wang et al. (2020a) also predicted the highest concentrations of acetone in the Arctic in June, July, August of around 8-12





nmol dm⁻³, in agreement with our measurements. Episodes of highest acetone concentrations tended to be observed during very brief episodes (1-3 h) and near land during the latter part of the cruise.

Acetone displays higher mean concentrations in sea ice covered waters (10.9 nmol dm⁻³) compared to ice-free waters (8.3 nmol dm⁻³) (*t*-test, *n1*=202, *n2*=61, *t* stat=2.5, *t* critical=1.6, p=0.01) (Fig. 9). Higher concentrations of acetone in partially sea ice covered ocean could be due to exposure of photolabile organic carbon from under the sea ice and the influence of sea ice on

light penetration depth, thus further supporting a dominant photochemical source of acetone during this cruise track.

### 4.3 Acetaldehyde

Most acetaldehyde depth profiles display a rapid decline in concentration from the surface to about 20 m, where it reaches a value nearly constant with depth (Figure 4, Figure 7). Many of the casts display higher concentrations of acetaldehyde at 30 cm compared to at 2 m, especially when those depths display different densities (Fig. 4). We note that the absolute acetaldehyde

concentrations presented here are uncertain due to an unquantified interference of $CO_2$ with the background of acetaldehyde (Supplement S2). The amount of $CO_2$ within the 60 m near the surface is not expected to vary drastically (Beaupré-Laperrière et al., 2020) and should thus not impact the shape of the acetaldehyde depth profiles, which are of value. Sharing some similarity to the acetone depth profiles, the rapid decline of acetaldehyde concentrations from the surface likely suggests a dominant light-dependent source near the surface of the water column, which is supported by previous studies (Dixon et al.,

2013; Mopper and Stahovec, 1986; Zhou and Mopper, 1997; Zhu and Kieber, 2019). In contrast to acetone, acetaldehyde almost never shows a homogenous profile within the mixed layer of ice-free casts. This may be because acetaldehyde lifetime in seawater is too short (a few hours (Dixon et al., 2013)) to be mixed homogenously, in contrast to acetone with its longer lifetime in seawater (5 to 55 days (Dixon et al., 2013)). These profiles from the sea ice zone are in contrast to previous measurements in the open ocean of the Atlantic where generally similar concentrations of acetaldehyde are observed at the

surface compared to below the mixed layer (Beale et al., 2013; Yang et al., 2014b). These Arctic profiles compare best to depth profiles nearer to land (Beale et al., 2015; Kieber et al., 1990; Zhu and Kieber, 2019), which also observe a rapid decline in concentration with depth, likely due to rapid light attenuation (Zhu and Kieber, 2019). This suggests that light penetration depth strongly influences the shape of acetaldehyde depth profiles. In the Arctic, light penetration depth is largely governed by freshwater input from melting sea ice. In seawater, it is thought that 7-53 % (Zhu and Kieber, 2019) or 16-68 % (Dixon et

al., 2013) of acetaldehyde is produced from photochemical activity. The remainder is likely produced in a light-dependent manner from biological activity (Davie-Martin et al., 2020; Halsey et al., 2017). As mentioned previously, the wavelengths of visible light required to produce acetaldehyde from biological activity penetrate to $\approx 40 - 50$ m (Massicotte et al., 2018) (Figure 7), while the wavelengths responsible for photochemical production are expected to penetrate only to 2-7 m (Tedetti and Semperv, 2006). Indeed, Zhu and Kieber (2019) model that about 90 % of the acetaldehyde photochemical production occurs

within the upper 4 and 23 m in coastal and open ocean waters respectively. Thus, it is likely that some of the acetaldehyde below the penetration depth of UV light is produced from biological activity. Crucially though, photochemical production, rather than biological production, appears to control the amount of acetaldehyde at the surface and thus available for air – sea





exchange. We might expect a peak in acetaldehyde concentration at the deep Chl *a* maximum if biological production was dominant, even if it only receives 3–10 % of the surface irradiance (Martin et al., 2010). Our observations generally show

lower concentrations below 30 m, suggesting that the main source of acetaldehyde in this area is probably photochemistry, rather than biological production. Additionally, the acetaldehyde casts from this deployment show remarkable consistency in profile shape, while Chl *a* (as an indicator for biological activity) was highly variable. This further supports that photochemical production in this area may be the dominant production process of acetaldehyde.

Mean (median) seawater acetaldehyde concentration was 3.7 (3.9) nmol dm$^{-3}$ (Figure 8). We reiterate that the acetaldehyde

concentration measurement is possibly biased due to uncertainty in the background value. Nevertheless, this mean concentration in the Arctic compares well with open ocean concentrations from the Atlantic (Beale et al., 2013; Yang et al., 2014b; Zhu and Kieber, 2019) and the Pacific (Kameyama et al., 2010) as well as measurements in shelf areas (Beale et al., 2015; Schlundt et al., 2017; Zhou and Mopper, 1997). There are episodes of remarkably high acetaldehyde concentrations (around 10 nmol dm$^{-3}$) during this cruise track. High biological (Burgers et al., 2020) and photochemical activity (Mungall et

al., 2017; Ratte et al., 1998) combined with 24 h daylight might have led to strong production of acetaldehyde during some periods of our study.

Excluding data at SIC=0, underway acetaldehyde displays a positive correlation with SIC, with an $R^2$ value of 0.29 (Fig. 9). Higher concentrations of acetaldehyde in partially sea ice covered ocean could be due to exposure of photolabile organic carbon from under the sea ice. The Arctic summertime is a hotspot for photochemical production of organic compounds

(Mungall et al., 2017; Ratte et al., 1998). The origin of organic carbon has previously been shown to strongly influence the production rate of these compounds (De Bruyn et al., 2011b), with unbleached, terrestrial organic carbon appearing to be more effective precursors (Zhu and Kieber, 2018) and dominant in this sampling area (Mungall et al., 2017).

## 4.4 Relationships between oxygenated VOCs

In this dataset, underway acetaldehyde and acetone correlate significantly ($R^2$ = 0.35, P<0.001, N=247). Underway

acetaldehyde and methanol also correlate significantly in this dataset ($R^2$ = 0.34, P<0.001, N=248), while the correlation between methanol and acetone is also significant ($R^2$ = 0.32, P<0.001, N=262). These correlations are significant, suggesting common sources of these compounds over this cruise track but their predictive qualities are quite poor. Yang et al. (2014b) have observed similar correlations during a transatlantic transect with $R^2$ values of 0.29 (acetaldehyde vs. acetone) and 0.25 (acetaldehyde vs. methanol). However, they did not observe a correlation between methanol and acetone during their

deployment. Likewise, Schlundt et al. (2017) observed correlations between acetone and acetaldehyde surface seawater with $R^2$ values around 0.5 in the South China/Sulu Sea. The correlation between acetone and acetaldehyde is likely due to common photochemical sources in this area. Similarly, acetaldehyde and acetone correlate with methanol, likely due to common light-dependent sources.

All three oxygenated VOCs (methanol, acetone and acetaldehyde) measured during this cruise generally display lower

concentrations during the first week of sampling, which corresponds to sampling the sea ice zone of the more marine-





influenced Davis Strait and Baffin Bay area. The slightly higher concentrations of these compounds nearer to land, i.e. in the channels of the Canadian Archipelago, may be related to terrestrial sources.

## 4.5 DMS

Stations with near full ice cover (75 to 90 % SIC) display highest concentrations of DMS within the 10 m near the surface
(Fig. 5a). This could be related to phytoplankton at the bottom of the sea ice seeding ice edge blooms, which are known to be sources of DMS (Levasseur, 2013). Stations with partial sea ice cover (20 to 50 % SIC) and ice-free stations (0 % SIC) (Fig. 5b and c) display higher concentrations of DMS at deeper depths (ca. 10-20 m), in part due to the establishment of a deeper stratified mixed layer (Figure 5b and Fig. 5c, Figure 7). DMS maxima below the mixed layer are sometimes accompanied by deep Chl *a* maxima, qualitatively similar to previous observations of DMS profiles in oligotrophic waters (Simó et al., 1997)
and the sea ice zone (Abbatt et al., 2019; Galí and Simó, 2010). Whether a DMS maximum occurs at the same depth as the deep Chl *a* maximum or not likely depends on the biological community composition (Galí and Simó, 2010; Levasseur, 2013). We generally observe similar concentrations of DMS at 2 m and at 30 cm.

The cruise mean DMS concentration was 1.42 nmol dm$^{-3}$, which is similar to the median concentration of 1.35 nmol dm$^{-3}$ (Figure 8). This is within the range of concentrations measured by Jarnikova et al. (2018) but lower than measurements by
Mungall et al. (2016) and Abbatt et al. (2019) in the same region also during July and August. Previous measurements generally show the lowest DMS concentrations before ice break up (Bouillon et al., 2002) and during the sea ice minimum (Luce et al., 2011; Motard-Côté et al., 2012).

There appears to be noticeable variability in surface DMS concentrations in the Arctic on both seasonal and inter-annual timescales (Collins et al., 2017). The seawater concentrations measured in Northern Baffin Bay during the cruise presented
here show remarkably good agreement with concentrations of approximately 1 nmol dm$^{-3}$ predicted by Galí et al (2019) based on a satellite algorithm. Their satellite algorithm suggests that the majority of the cruise sampling presented here has been carried out after peak DMS concentrations in this area (Galí et al., 2019). This could be the reason why other investigators have recently measured higher concentrations in this area than we report here (Abbatt et al., 2019; Mungall et al., 2016).

Jarnikova et al. (2018) observed higher surface DMS concentrations near strong gradients in SIC. The sea ice zone (Levasseur,
2013) and the ice edge of the Canadian Arctic Archipelago (Abbatt et al., 2019) have previously been identified as strong sources of DMS. Excluding data collected in open water, we generally observe slightly higher surface DMS concentrations at higher sea ice concentrations (Fig. 9), however no significant correlation could be observed. This could be because the relationship between DMS and SIC is more complex and highly dependent on the biological settings. Higher concentrations in partial sea ice cover could be in part due to production of DMSP induced by large shifts in salinity and temperature, which
is further metabolised into DMS (Levasseur, 2013; Wittek et al., 2020).



### 4.6 Isoprene

The stations with highest ice cover (75 to 90 %) display the highest isoprene concentrations at the surface, and the concentrations decrease gradually with depth over the upper 50 m (Fig. 6a). At lower SIC (20 to 50 %) and in ice-free casts (0 % SIC), the highest isoprene concentrations often occur below the surface, sometimes coinciding with the deep Chl $a$ maximum

(Fig. 6b and c, Fig. 7). Previous depth profiles from the open ocean showed that isoprene frequently displays a subsurface maximum, which can be located either at, above or below the Chl $a$ maximum and can be related to the oxygen maximum (Booge et al., 2018; Hackenberg et al., 2017; Tran et al., 2013). In the casts here, isoprene often displays a subsurface maximum at the deep Chl $a$, which is characteristically located just below the mixed layer (Martin et al., 2010). This frequently coincides with higher oxygen concentrations at the same depth, suggesting that gases produced at this depth from biological activity are

not efficiently vented to the atmosphere (Fig. 7). As suggested also by Hackenberg et al. (2017), the frequent deep isoprene maximum confirms that there is substantial isoprene production at depths of 10 m or deeper, also in the sea ice zone. Lower concentrations of isoprene in the mixed layer compared to at the deep Chl $a$ maximum are likely in part due to ventilation to the atmosphere. Some of the casts display higher concentrations of isoprene at 30 cm compared to at 2 m. Similarly to methanol, we speculate that this could be due to lethal levels of UV light leading to cell lysis or other biological processes near the

surface. This concentration gradient is likely preserved due to differences in density.

The mean isoprene concentration was 0.063 nmol dm$^{-3}$, which is similar to the median concentration of 0.059 nmol dm$^{-3}$ (Figure 8). This is suggesting a relatively normal distribution of isoprene concentrations during this deployment. Overall these isoprene concentrations appear about twice as high compared to other open ocean measurements (Hackenberg et al., 2017; Ooki et al., 2015). Measurements from this cruise compare better to measurements in very biologically productive areas (Baker

et al., 2000; Matsunaga et al., 2002; Shaw et al., 2010b) and in coastal regions (Baker et al., 2000; Hackenberg et al., 2017; Ooki et al., 2015, 2019; Shaw et al., 2010b).

Previous authors have suggested Chl $a$ as an indicator of surface isoprene concentrations (Hackenberg et al., 2017; Ooki et al., 2015; Rodríguez-Ros et al., 2020), as isoprene is produced by a range of phytoplankton (Shaw et al., 2010b). However, we observe only a very weak positive correlation between underway isoprene and Chl $a$. This could be because most of the

phytoplankton bloom and isoprene production occurs under the ice, before it is sampled by the ship (Ahmed et al., 2019). The slope and intercept of regressing underway isoprene vs. Chl $a$ is 0.024 and 0.059 respectively ($R^2$ = 0.04, p=0.001, N=222). Hackenberg et al. (2017) and Ooki et al. (2015) have suggested a positive correlation between isoprene and sst from open ocean measurements. Contrary to those results, we actually observe a negative correlation between isoprene concentrations vs. sst during this cruise. The slope and intercept of regressing underway isoprene vs. sst is -0.0030 and 0.0641 respectively ($R^2$

= 0.12, P=0.01, N=222), suggesting highest isoprene concentrations in colder waters. Over this cruise track, some of the variability in surface isoprene could be explained by SIC. Excluding data collected without sea ice cover, the slope and intercept of regressing underway isoprene vs. SIC is 0.00024 and 0.057 respectively ($R^2$ = 0.19, P=0.001, N=42). Higher isoprene concentrations at greater SIC could be due to ice edge blooms and higher biological production (indicated by Chl $a$)





in partial sea ice cover or in the recently ice uncovered water column. These correlations of isoprene suggest a unique influence

of seasonal sea ice melt on isoprene concentrations. By inference, parametrizations that predict surface isoprene concentrations as a function of Chl *a* and sst (Ooki et al., 2015; Rodríguez-Ros et al., 2020)), developed based on open ocean measurements, might not be applicable to the sea ice zone.

These depth profiles and underway data represent measurements at different times and locations. Therefore, differences are

possibly not only due to sea ice coverage but could also be due to the oceanography of the area (McLaughlin et al., 2004). We recognise that sea ice is a very heteogenous environment with respect to ice thickness (Hayashida et al., 2020), the presence of melt ponds (Gourdal et al., 2018; Park et al., 2019), and types of sea ice (e.g. first year vs. multiyear ice (Lizotte et al., 2020). This heterogeniety likely leads to very dfferent biogeochemistry, affecting trace gas cycling. It is worth noting that the analysis presented here does not explicitly take into consideration this variability, which is worthy of future research.

**5 Air – sea fluxes**

Air – sea fluxes are calculated using the Liss and Slater (1974) two layer model. The equations used in our calculation are laid out in detail in Wohl et al. (2020). Briefly, methanol, acetone and DMS fluxes are computed using the waterside transfer velocity by Yang et al. (2011) and the airside transfer velocity by Yang et al. (2013a). Isoprene fluxes are computed using the waterside transfer velocity from Nightingale et al. (2000) and the airside transfer velocity from Yang et al. (2013a). Schmidt

numbers for methanol, acetone, acetaldehyde and DMS were calculated following Johnson (2010), while the Schmidt number of isoprene was calculated using the equation from Palmer and Shaw (2005). Solubilities in seawater listed in Wohl et al. (2019) were calculated as a function of seawater temperature. For methanol and acetone, the solubility determined in Wohl et al. (2020) was used. These solubilities were used to calculate mean saturation and seawater concentration at equilibrium with the atmosphere.

A detailed discussion of the effect of sea ice on the gas transfer velocity is beyond the scope of this study. Thus, we simply scale the air – sea flux linearly to the open water fraction (using AMSR2 derived SIC), treating the sea ice as a barrier to air – sea exchange on this regional scale as recommended by Butterworth and Miller (2016) from measurements in the Antarctic and Prytherch et al. (2017) from measurements in the Arctic. We further compute surface saturations as:

$$s\,/\,(\%) \;=\; \frac{C_w/(nmol\,dm^{-3})}{C_a/(nmol\,dm^{-3})\,H/(1)}$$

where $C_w$ and $C_a$ are the concentrations in water and air respectively and H is the dimensionless water over liquid form of the Henry solubility. Saturations below 100 % indicate oceanic undersaturation, while negative fluxes indicate ocean uptake i.e. air-to-sea flux. We can also evaluate the state of saturation from an air perspective. The equilibrium gas phase mixing ratio calculated for methanol and acetone is the gas phase mixing ratio that is at equilibrium with the measured seawater concentration. Ambient mixing ratios below these values suggest oceanic outgassing.





Atmospheric mixing ratios of these gases were not measured during this campaign. Therefore, we use a constant mixing ratio based on literature values to calculate fluxes and saturations. For methanol and acetone, we assume a constant mixing ratio of 0.3 and 0.4 ppbv respectively as measured by Sjostedt et al. (2012). Due to the lack of ambient air measurements during this cruise, the estimated fluxes of methanol and acetone are rather uncertain (i.e. air/sea concentration difference highly sensitive to atmospheric concentration, Yang et al. (2014a)). Ambient air concentrations of isoprene and DMS generally have a very

small influence on the calculated air – sea flux due to the low solubility/large supersaturation (Baker et al., 2000; Matsunaga et al., 2002) and the compounds' short lifetimes in the atmosphere (Medeiros et al., 2018). Thus, we assume the ambient air concentrations of isoprene and DMS to be zero in our flux estimates. Acetaldehyde fluxes were not computed due to the uncertainty in absolute acetaldehyde concentrations related to uncertain background corrections (Supplement S2). We use our underway measurements from 3-4 m depth to calculate air – sea fluxes, thus not accounting for the sometimes slightly higher

seawater concentrations of some compounds observed at some stations at 30 cm depth.

We calculate a mean flux of methanol into the ocean of -3.3 $\mu$mol m$^{-2}$ d$^{-1}$ and a mean saturation of 22 %. The equilibrium gas phase mixing ratio was 0.06 ppbv, suggesting that the flux of methanol was most likely consistently into the ocean. Direct flux measurements of methanol in the Labrador Sea and during a trans-Atlantic crossing typically report -20 to -10 $\mu$mol m$^{-2}$ d$^{-1}$, but similar saturations of around 20 % (Yang et al., 2013a, 2014a). It is likely that the calculated methanol fluxes from this

cruise are lower due to the low wind speeds during this cruise (Figure 10) and sea ice acting as a barrier to air – sea exchange in this calculation. Oceanic uptake of methanol is probably due to the extremely high solubility of this compound combined with the relatively high air mixing ratios, characteristic of marine air in the Northern Hemisphere (Bates et al., 2021; Galbally et al., 2007).

The mean flux of acetone during this deployment is calculated to be into the ocean at -3.3 $\mu$mol m$^{-2}$ d$^{-1}$, while the mean

saturation is 27 %. The equilibrium gas phase mixing ratio is calculated as 0.10 ppbv, which suggests that the acetone flux was largely into the ocean with episodes of outgassing possible. This mean flux is within the range of previous acetone flux measurements in the open ocean of typically around 10 to -10 $\mu$mol m$^{-2}$ d$^{-1}$ (Schlundt et al., 2017; Taddei et al., 2009; Yang et al., 2014b, 2014a). Acetone was most likely undersaturated due to the relatively low seawater concentrations and high atmospheric mixing ratios that are characteristic of marine air in the Northern Hemisphere (Galbally et al., 2007; Wang et al.,

2020a).



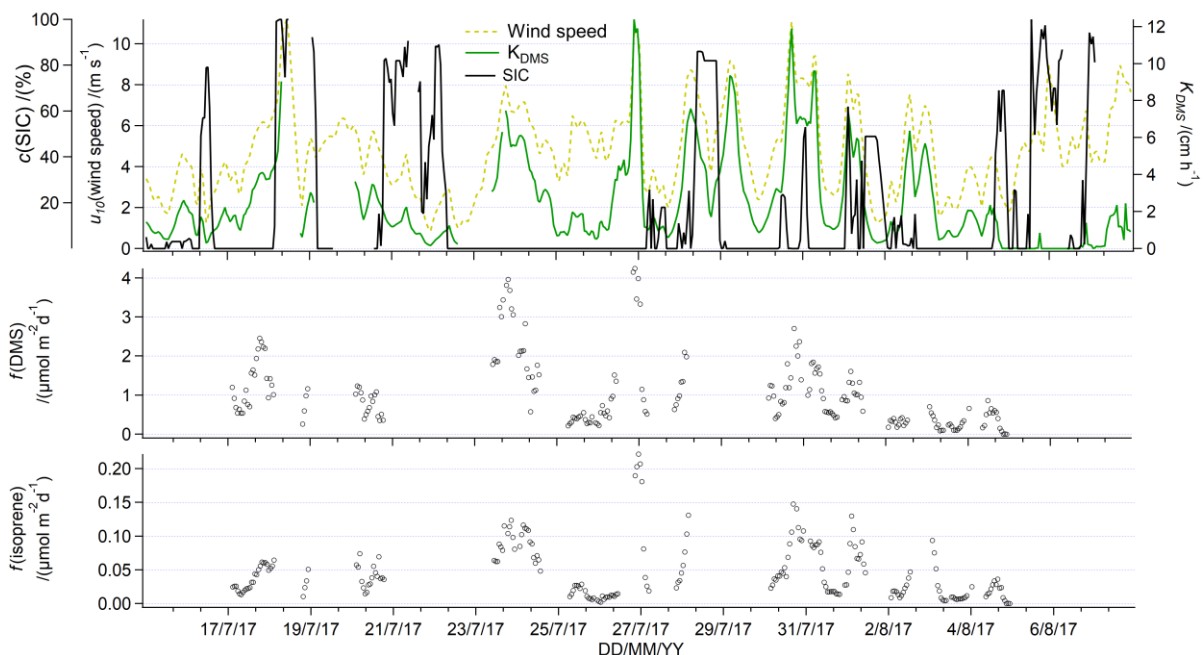

**Figure 10 (a) Timeseries of underway sea ice concentration, wind speed at 10 m and calculated air sea exchange velocity. Short gaps are due to gaps in the recording of the underway sea surface temperature. (b) Calculated underway fluxes of isoprene.**

In Fig. 10, we show wind speed, sea ice concentration and estimated DMS transfer velocity, as well as the fluxes of DMS and isoprene.

A cruise mean DMS flux of 1.05 µmol m$^{-2}$ d$^{-1}$ has been calculated while the median is only 0.84 µmol m$^{-2}$ d$^{-1}$ (Figure 10). This is within the range of directly measured DMS fluxes in the Labrador sea in October/November of 1.5 µmol m$^{-2}$ d$^{-1}$ (Kim et al., 2017) or other calculated fluxes in the Canadian Arctic Archipelago of 0.2-12 µmol m$^{-2}$ d$^{-1}$ in July/August (Mungall et al.,

2016). During the campaign presented here, the highest fluxes of DMS of around 5 µmol m$^{-2}$ d$^{-1}$ were observed on 24/07 and 27/07 in ice-free conditions: These periods were marked by moderate wind speeds and near cruise average DMS seawater concentrations. They were both located in the northern Baffin Bay area. Previous modelling studies found large marine contributions to atmospheric DMS mixing ratios in this sampling area from the Baffin Bay (Mungall et al., 2016).

The mean flux of isoprene during this deployment was 0.047 µmol m$^{-2}$ d$^{-1}$, while the median was 0.033 µmol m$^{-2}$ d$^{-1}$ (Figure

10). These estimates are comparable to previous direct measurements of isoprene fluxes in the Labrador sea of on average 0.0718 µmol m$^{-2}$ d$^{-1}$ (Kim et al., 2017) or other calculated fluxes in open ocean (Broadgate et al., 1997; Matsunaga et al., 2002). It is surprising that the relatively high isoprene concentrations measured throughout this cruise track did not lead to higher fluxes. Relatively low fluxes of isoprene despite high seawater concentrations are due to the low wind speeds from this cruise and sea ice acting as a barrier to air sea exchange in our calculation. The mean isoprene flux is higher than the median, which

suggests that isoprene fluxes are dominated by episodic emissions related to biological productivity and wind and sea ice driven air – sea exchange. For example, as with DMS, highest emissions of isoprene of up to 0.2 µmol m$^{-2}$ d$^{-1}$ were observed

in the northern Baffin Bay area on 24/07 and 27/07. These episodes were marked by some of the highest wind speeds of the campaign (10 m s$^{-1}$) and very low sea ice coverage. In fact, over this cruise track, isoprene and DMS fluxes correlate significantly. The correlation of the isoprene flux as a function of the DMS flux gives a slope of 0.038 and an intercept of
0.006 ($R^2$ = 0.68, P<0.001, N=212). This suggests that DMS and isoprene tend to be emitted together in the same, ice-free locations even if their underway seawater concentration did not significantly correlate. This is largely because fluxes in the sea ice zone tend to be controlled by wind speed and SIC. Generally higher DMS and isoprene dissolved concentrations at higher SIC and highest emissions at low SIC/high winds suggests that both of these gases are produced at high SIC and released to the atmosphere when the ice retreats or in ice-free conditions. These concurrent peak DMS and isoprene emissions will
impact the overlying atmosphere.

To calculate the lifetime of isoprene in seawater relative to air – sea exchange, we assume an approximate mean mixed layer depth of 15 m (estimated from Fig. 2 and Fig. 7), which is divided by the cruise mean isoprene transfer velocity (2.51 cm h$^{-1}$). This gives a mean lifetime of isoprene with respect to air – sea exchange of 24 days. Previous authors have estimated this to be 7 (Palmer and Shaw, 2005) or 10 (Booge et al., 2018) days. This implies that ventilation to the atmosphere is a smaller sink
of isoprene during this cruise compared to the open ocean, possibly due to the low wind speeds characteristic of summer in this area (McLaughlin et al., 2004) and sea ice acting as a barrier to air – sea exchange. Reduced ventilation may also contribute to the relatively high isoprene seawater concentrations observed in the sea ice zone compared to open ocean measurements.

## 6 Conclusion

This paper presents depth profiles and underway seawater measurements of methanol, acetone, acetaldehyde, DMS and
isoprene in the marginal sea ice zone. The measurements were taken in the Canadian Arctic Archipelago during July/August 2017, i.e. during Arctic summer/sea ice melt. To the best of our knowledge, these represent the first measurements of seawater concentrations of methanol, acetone, acetaldehyde and isoprene in the Canadian Arctic Archipelago. The underway measurements are also used to calculate air – sea fluxes.

To synthesize the observations, a summary of the effect of different sea ice concentrations on dissolved gas distributions is
provided here as a schematic (Figure 11).



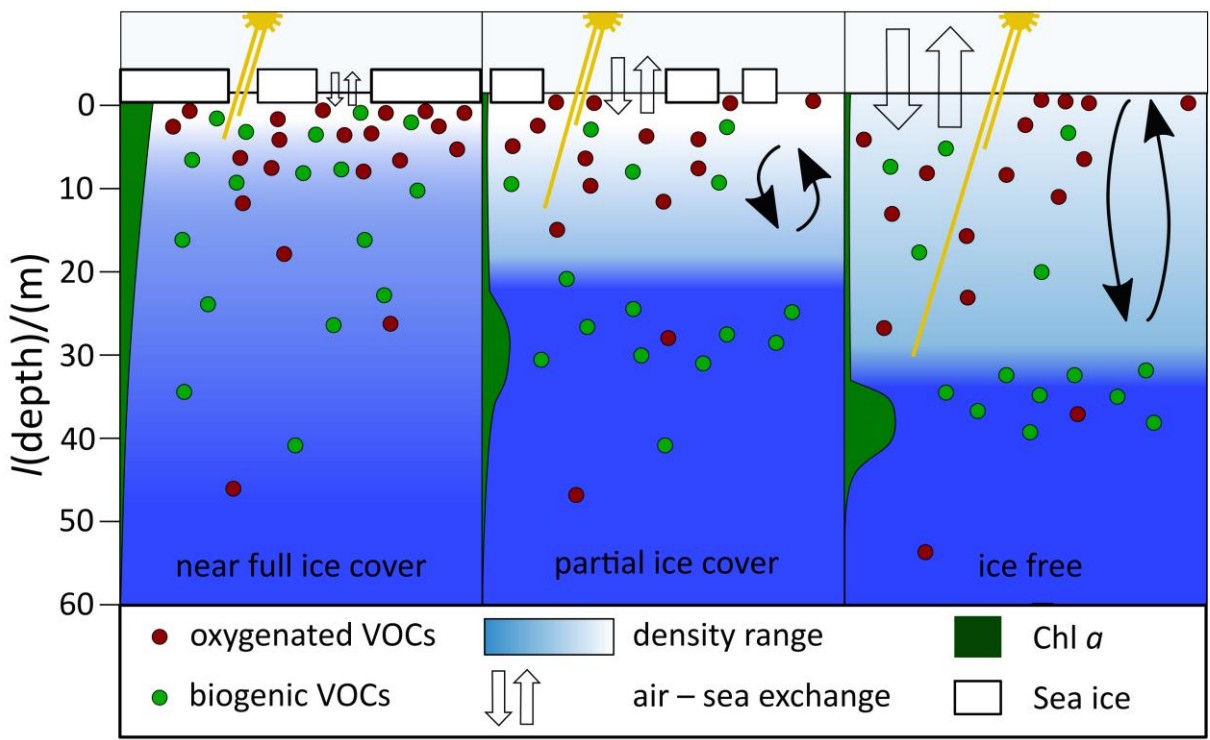

**Figure 11 Schematic summarising the impact of seasonal sea ice melt on dissolved gas concentrations. Methanol, acetone and acetaldehyde = "oxygenated VOCs", DMS and isoprene = "biogenic VOCs". The shorter light beam represents UV rays, while the longer one represents PAR.**

For ease of illustration and discussion, here we group methanol, acetone and acetaldehyde as "oxygenated VOCs" and DMS and isoprene as "biogenic VOCs".

Oxygenated VOCs tend to display the highest concentrations near the surface and do not display a subsurface maximum. They often display slightly higher concentrations at 30 cm compared to 2 or 5 m if a surface stratified layer is present. Underway methanol concentrations display a large range in concentration in the sea ice zone and generally higher concentrations near the

surface in ice-free waters. This appears to be consistent with rapid biological cycling (Mincer and Aicher, 2016; Sargeant et al., 2016) where the sources and sinks are at times decoupled. Acetone and acetaldehyde concentrations decline rapidly from the surface to reach a constant value at about 5 m in near-full ice cover and at about 20 to 30 m in partial ice cover (less than 50 % SIC) and ice-free waters. This is probably due to deeper UV light penetration at lower SIC and increased mixing as the mixed layer forms. Higher concentrations in the top 5 m of the water column support a dominant photochemical, UV light

dependent, source of these compounds in the sea ice zone as wavelengths required for biological production would be expected to penetrate deeper. Despite obvious sources in seawater, we calculate that the sea ice zone is highly undersaturated in methanol and acetone, leading to net ocean uptake of these gases.





The biogenic VOCs, DMS and isoprene, behave differently to the oxygenated VOCs as they sometimes display a subsurface maximum and their concentrations are generally similar between the depths of 30 cm and 5 m. In casts with high ice cover (75

to 90 % SIC), we observe gradual declines of DMS and isoprene concentrations from the surface down to about 50 m. In partial ice cover and in ice-free conditions (less than 50 % SIC), DMS and isoprene concentrations in the mixed layer are more homogenous. Many of these casts display a deep isoprene maximum, but only sometimes a deep DMS maximum. Isoprene and DMS surface concentrations were both higher at higher SIC. Surface isoprene concentrations correlated more strongly with SIC than with sst or Chl *a*, suggesting that isoprene concentrations in the sea ice zone are controlled by different processes

than in the open ocean. DMS and isoprene fluxes were tightly correlated, even if their seawater concentrations were not. Greatest emissions of the biogenic VOCs were observed in ice-free areas. Our calculation of the lifetime of isoprene relative to air – sea exchange suggests that sea ice leads to reduced ventilation of isoprene and thus a longer lifetime of dissolved isoprene in the sea ice zone compared to open ocean. This contributes to relatively high seawater isoprene concentrations in the sea ice zone.

Taken together, our observations suggest that sea ice concentration exerts a strong influence on dissolved VOCs via an interplay between physical drivers (e.g. mixing, seasonal stratification, light penetration, wind speed) and biogeochemistry. These measurements and insights improve our understanding of the cycling of these gases in the polar oceans. The air – sea fluxes of DMS and isoprene will be helpful for improving estimates of the aerosol budgets in the changing Arctic. Simultaneous emission of DMS and isoprene suggests that they are part of the cocktail of gases released into the atmosphere from the recent

ice uncovered water column. Similarly, the air – sea fluxes of methanol and acetone will be helpful to assess the impact of these oxygenated VOCs on the oxidative capacity of the atmosphere and thus the lifetime of atmospheric pollutants and methane in the Arctic atmosphere.

With further sea ice loss predicted for a changing Arctic, we speculate that this is going to lead to higher emissions of biogenic VOCs in the future. For example sea ice loss over the last approximately 20 years has led to increased emissions of DMS (Galí

et al., 2019) which also affected aerosol concentrations in summer (Sharma et al., 2012). Due to increased sea ice loss and the subsequent increase in air-to-sea flux, we speculate that the Arctic Ocean will be a bigger sink for oxygenated VOCs, thereby reducing their atmospheric concentration, which affects oxidative capacity.

Further observations should focus on year-round observations at the same site to reduce some of the variability in the data due to heterogenous sea ice biogeochemistry and local oceanography.

**8 Data availability**

Data has been submitted to Polar Data Catalogue (https://www.polardata.ca/pdcsearch/), where the CCIN Reference number is 13249. A DOI will be issued in due course and included in the published manuscript. The CCIN number makes the dataset searchable for the period of the peer review process.
## 9 Author contribution

MY, AEJ, WTS and PDN conceptualised the project. CW carried out the measurements and data analysis. BE provided crucial resources for the project. BB provided the wind speed measurements. CW wrote the manuscript with input from all co-authors.

## 10 Competing Interest

The authors declare that they have no conflict of interest.

## 11 Acknowledgements

These measurements were made possible through a large range of collaborations. We are thankful to Mohamed Ahmed, Dave Capelle, Tonya Burgers, Douglas Collins, Jonathan Abbatt and Martine Lizotte for logistical support. Many thanks to Emily Alcock for her support during the write up. Final thanks go to the excellent crew of the CCGS *Amundsen* and the chief scientists Jean-Éric Tremblay and Martine Lizotte.

Some of the data presented herein were collected by the Canadian research icebreaker CCGS Amundsen and made available

by the Amundsen Science program, which is supported by the Canada Foundation for Innovation Major Science Initiatives Fund. The views expressed in this publication do not necessarily represent the views of Amundsen Science or that of its partners.

We thank the Institute of Environmental Physics, University of Bremen for the provision of the merged MODIS-AMSR2 sea-ice concentration data at https://seaice.uni-bremen.de/data/modis_amsr2 (last access 11/01/2019).

## 12 Financial support

This work was supported by the Natural Environment Research Council through the EnvEast Doctoral Training Partnership (grant no. NE/L002582/1) and by the UK Department for Business, Energy and Industrial Strategy (United Kingdom & Canada Arctic Partnership: 2017 Bursaries Programme awarded to MY). Financial support was provided to BE by the National Sciences and Engineering Research Council of Canada. This work is a contribution to ArcticNet, a Network of Centres of
Excellence Canada.

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
