# Peer review of "Sea ice concentration impacts dissolved organic gases in the Canadian Arctic"

_Biogeosciences, 2021_

## Author Comment (AC1)

**Reply to Reviewer 1 comments for: "Sea ice concentration impacts dissolved organic gases in the Canadian Arctic" by Charel Wohl et al.**

Many thanks to the reviewer for taking the time to thoroughly review the manuscript and provide constructive comments. The reviewer provided thought provoking comments, which has helped us to improve the manuscript. Please see our responses below. Reviewer comments are in normal font and author's replies can be found in italic.

**General Comments**

As we know, the sea ice paly an important role in influence the dynamic of biological activity and photochemistry in polar oceans, and subsequently, indirectly impact the production and release of OVOCs and biogenic gases. The author clear presented the temporal and spatial distributions of gases associated with the sea ice concentrations. However, to make the conclusion that the sea ice concentration impacts the gases is not preciseness. They did not perform a timeseries observation at a stable station to investigate the influence of sea ice dynamic to seawater gases. Might be the title "characteristic of dissolved gases in marginal sea ice area in Canadian Arctic" is more suitable.

On the other hand, the authors should also make some description in the dynamic of nutrients and phytoplankton activities from previous reports. To my knowledge, in the July or August, the Arctic Ocean in Baffin Bay is possible flourishing with high phytoplankton biomass (Bloom). If the nutrients were enough to support the growth of phytoplankton, it would be easily to observed the bloom in the marginal sea ice area. However, along the cruise track, both biogenic gases and Chl a indicated low values except some period like July 27-28. Please also check the satellite Chl a data from ocean color website. Then, we can know how the phytoplankton growth in the whole Bay. The seawater DMS levels < 3 nmol L-1 were not high compared with previous studies. If there is no problem with the measurement method, the phenomena should be noticed. Might be the phytoplankton bloom in early July or June consumed the surface nutrients. Thus, to know the profile nutrients information is very important to explain the data.

*Thank you for suggesting a different title. Indeed, to better understand and be more conclusive, future research should focus on making year-round observations at the same station. We added this idea to the end of the manuscript, under future research. Most of the analysis in the paper focusses on how VOC concentrations change with sea ice concentration in space, which we take to be a proxy for the seasonal influence of sea ice. While this sampling strategy wasn't Lagrangian, we prefer to stay with the current title as the word 'impacts' does not necessarily imply direct causation.*

*Indeed, the DMS concentrations from this cruise are lower than previous measurements in this region at a similar time of year. In the manuscript, we discuss that our DMS concentrations are lower than previous measurements. We explain that peak DMS concentrations probably occurred before our sampling campaign, a hypothesis supported by satellite inferred DMS estimates by Galí et al. (2018). Also Collins et al. (2017) found high interannual variability in DMS concentrations, despite approximately sampling at the same time and same location over two years. DMS in the Arctic is also spatially very heterogenous, so it is also possible that our cruise track did not cover any DMS hotspots. Nutrient concentrations from this cruise have only been made available very recently and a brief discussion has been included in the revised manuscript.*

*The following changes have been made to the manuscript in regards to this comment made by the reviewer:*

*In Sec. 2.1, the following sentence has been added:*

*Inorganic nutrient measurements (nitrate) were carried out as described in Randelhoff et al. (2019).*

*Nitrate measurements have been added to the casts presented in Figure 7.*

*When discussing the underway auxiliary data (Sec. 3.2), we added the following sentence:*

*Nitrate concentrations at 5 m ranged between 0 and 0.7 µmol dm$^{-3}$ suggesting that the phytoplankton bloom sampled here is very advanced as nutrients near the surface are deplete.*

*DMS discussion (Sec. 4.5) added:*

*Low surface nitrate concentrations measured during this cruise also suggest that the sampling presented here has been carried out after peak phytoplankton growth.*

*Satellite Chl a data was overall not as conclusive due to poor satellite coverage and spatial heterogeneity of the spring bloom. But overall it shows higher Chl a concentrations at the beginning of July compared to the beginning of August, further supporting that most sampling presented here occurred after the peak bloom phase.*

**Some minor points**

Line 63, There is no Zhang et al., 2019 in the references. Please also check the whole manuscript.

*The reference list has been updated and this has been corrected.*

Figure 1, if you can present the real sea ice cover data (from https://seaice.uni-bremen.de/data/amsr2/), it would be clear.

*The sea ice covered area is approximately indicated for illustration purposes as a shaded area due to the dynamic nature of sea ice cover and difficulties of conveying this information for a month-long deployment. The approximate location of the sea ice edge is based on the average sea ice concentration for the whole cruise duration using AMSR2 satellite data.*

*The figure description has been updated to make this clearer.*

Line 111, Chl a measured by the sensor might not precise. Is there any biologist do the measurement Chl a through filtering the water? You can use this data if you have choice.

*Unfortunately no direct HPLC measurements of the chlorophyll a pigment were made during this cruise. Ship technicians compared the fluorescence measured underway and the CTD mounted sensor. They find a linear correlation of 1:1 between both sensors and estimate that the accuracy of the fluorescence measurement is on the order of 0.1 µg mol$^{-1}$. Please do also note that apart from the correlation between isoprene and Chl a, most discussion of Chl a relies on relative, rather than absolute concentrations.*

Line 125, why not use the AMSR2 data in figure 1?

*Briefly, the sea ice edge data is based on an AMSR2 average, but only shown here as an illustration and thus approximate location.*

*This has been updated in the figure description.*

Line 147-149, what is the clear number of the difference. Is it significant?

*The difference between CTD and underway due to this acetaldehyde contamination was highly variable. We observed on the order of 500 nmol dm$^{-3}$ (approximately a factor of 100) higher concentrations in the CTD. This led us to label these clearly as a contamination.*

Why you use the 20%-50% SIC in figures for discussion? You should explain that in Method. Is there any define for heavy sea ice area or ice-free area?

*The casts have been grouped in panels by SIC. The grouping is based on the following definitions; in remote sensing, an ice-free area is generally considered to display ice coverage of less than 15 % (Wang et al., 2020b) and ice breakup has been defined by Ahmed et al. (2019) as the time the sea ice concentration changes from above to below 90 %.*

*Based on these few definitions, sampled stations, and corresponding SIC, stations were grouped as (a) 75-90 % near full ice cover/during ice break up, (b) 50-20 % partial sea ice coverage and (c) 0-15 % ice-free. No stations were sampled with SIC 70-50 %.*

*This has been added to the manuscript.*

For me, it is difficult to read the figure 2-6, where is the station numbers? I do not know where the stations along the cruise track. Please mark it if it is possible. Or use the date to indicate it?

*We decided to use the date to indicate this and updated figure 2-6. This greatly improved the figure in our opinion. Thank you for your input.*

Line 482, as you presented that you did not measure the atmospheric gases, the flux calculation for those gases with high levels in atmosphere by using a constant value seems bring large uncertainty. The authors should pay attentions to make the conclusion of "source or sink". If there is any other published paper calculate by the same method, you can cite those papers to let the readers know that it is reasonable.

*Methanol and acetone fluxes were computed using published atmospheric measurements at a similar time of year and place, this is similar to methods used by Beale et al. (2015). Indeed, this brings about some uncertainty in the calculated flux and this is acknowledged in the text. However, methanol and acetone were highly undersaturated (mean cruise saturation 22 and 27 % respectively). Thus this should not hugely influence our conclusion that the sea ice zone is a sink to those gases. Additionally, we present the equilibrium gas phase mixing ratio, which should allow readers to assess the flux direction independently.*

*This additional explanation has been added to the manuscript.*

Figure 10, the caption is unclear.

*Thank you for pointing this out. The caption has been updated to read:*

*Figure 10 (a) Timeseries of underway sea ice concentration, wind speed at 10 m and calculated air sea exchange velocity for DMS. Short gaps in the timeseries of the air sea exchange velocity of DMS are due to gaps in the recording of the underway sea surface temperature. (b) Calculated underway fluxes of (b) DMS and (c) isoprene.*

Line 576, replace the "Greatest" with "higher" or ? Higher emissions of biogenic VOCs were observed in ice-free areas than those with heavy SIC. The value of flux is not significant with DMS or isoprene. Please also check the whole manuscript.

*We decided to update the manuscript and use the sentence suggested by the reviewer.*

---

## Author Comment (AC2)

**Reply to Reviewer 2 comments for: "Sea ice concentration impacts dissolved organic gases in the Canadian Arctic" by Charel Wohl et al.**

Many thanks to the reviewer for taking the time to thoroughly review the manuscript and provide constructive comments. The reviewer provided thought provoking comments, which has helped us to improve the manuscript. Please see our responses below. Reviewer comments are in normal font and author's replies can be found in italic.

**General Comments**

In this paper, the authors present a suite of dissolved gases measurements conducted along a three-week transect on the east side of the Canadian Arctic in July-August 2017. This data set gives a rare representation of the vertical and horizontal distribution of methanol, acetone, acetaldehyde, dimethyl sulfide and isoprene in the marginal ice zone in this part of the Arctic. The sampling protocols and the analytic technics are well described and appropriate. The results are well discussed and lead to interesting hypotheses regarding the controlling factors of this gases and how they are influenced by the presence of sea ice. The interpretation of the data is slightly too speculative in few instances (see specific comments), but in general, very convincing owing the excellent grasp the authors have on the literature. Overall, this is an interesting and well written paper.

Here I would like to share with the authors two points deserving attention. First, the area covered during the cruise is very large and encompasses different water masses, water circulation patterns, and marine ecosystems. Different 'regions' have been obviously sampled and pooling all the results together may be misleading. I am not proposing to change the way the results are presented (i.e. figures), but the potential importance of the characteristics of the three main regions (West Baffin Bay, Smith Sound, Lancaster Sound) on the gases measured should be mentioned in the Discussion. Second, the effect of ice edges on the biogeochemistry of the adjacent waters is very much influenced by the water circulation. Water masses moving out, in, or along ice edges will have different biological and chemical characteristics. This should be taken into account when comparing the ice edges sampled west of Baffin Bay, Smith Sound, and Lancaster Sound. The authors touched that aspect when referring to the recent work my Lizotte et al. (2020), but do not discuss how this may influence their ice edge results.

*Many thanks to the reviewer for their input. We agree that a discussion of different regions and the effect of ice edges and water circulation on these gases would be helpful. Following the suggesting by the reviewer, we have divided the cruise track in three sections according to the regions suggested by the reviewer. We computed mean seawater concentrations and fluxes for each of these sections. A table displaying mean concentrations for each of the sections has been added to the supplementary material. Some discussion has been added to explain the differences in mean concentrations in light of the dominant seawater currents in the area and the location of the ice edge.*

*The following descriptive text has been added to the manuscript:*

*Sect. 3.2:*

*To investigate the effect of sea ice edges and water mass circulation on the surface concentrations of these compounds, we calculate mean underway surface concentrations measured in different sampling areas; West Baffin Bay (17.07-23.07), Smith Sound (23.07-31.07) and Lancaster Sound (31.07-07.08). These areas were chosen as they divide the cruise track in three equally representable sections and allow to comment on the effect of water circulation relative to sea ice edges/ice bridges. Means and standard errors of dissolved gas concentrations and some auxiliary data are presented in the supplementary material (Supplement S3). Sea ice bridges are often observed north of Smith Sound and east of Lancaster Sound (Lizotte et al., 2020; McLaughlin et al., 2004). At the same time, surface waters flow southwards and westwards from these sea ice bridges (McLaughlin et al., 2004; Münchow et al., 2015). This makes Smith Sound and Lancaster Sound ideal locations to sample downstream of an ice edge.*

*The following table has been added to the Supplementary material S3:*

**Table S1: Mean and standard error (std err) of the underway seawater concentrations of dissolved gases and some auxiliary data. Means are presented for three sections of the cruise. Full section name and sampling period are stated here; West Baffin Bay (West BB)) (17.07-23.07), Smith Sound (23.07-31.07) and Lancaster Sound (31.07-07.08).**

| | West BB | | Smith Sound | | Lancaster Sound | |
|---|---|---|---|---|---|---|
| | mean | std err | mean | std err | mean | std err |
| $c$(methanol)/(nmol dm$^{-3}$) | 23 | 2 | 46 | 3 | 41 | 3 |
| $c$(acetone)/(nmol dm$^{-3}$) | 3.4 | 0.3 | 10.8 | 0.5 | 11.7 | 0.7 |
| $c$(acetaldehyde)/(nmol dm$^{-3}$) | 1.1 | 0.4 | 5.5 | 0.2 | 6.0 | 0.2 |
| $c$(DMS)/(nmol dm$^{-3}$) | 1.61 | 0.06 | 1.59 | 0.06 | 1.00 | 0.04 |
| $c$(isoprene)/(nmol dm$^{-3}$) | 0.057 | 0.005 | 0.062 | 0.003 | 0.066 | 0.003 |
| $f$(DMS)/($\mu$mol m$^{-2}$ d$^{-1}$) | 1.12 | 0.08 | 1.44 | 0.11 | 0.52 | 0.05 |
| $f$(isoprene)/($\mu$mol m$^{-2}$ d$^{-1}$) | 0.041 | 0.003 | 0.065 | 0.005 | 0.029 | 0.003 |
| $T$(sst)/(°C) | 0.69 | 0.15 | 1.86 | 0.13 | 1.13 | 0.12 |
| $c$(sss)/(1) | 30.4 | 0.07 | 30.1 | 0.8 | 28.0 | 0.2 |
| $c$(Chl $a$)/(mg m$^{-3}$) | 0.73 | 0.07 | 0.26 | 0.04 | 0.18 | 0.02 |
| $c$(SIC)/(%) | 21 | 3 | 8 | 1 | 20 | 2 |

*A little bit of discussion has been added in regards to this for each compound. In square brackets we share sentences that have not been added as response to this reviewer comment, but have been included here for context.*

*Sec. 4.1 Methanol*

*[Underway methanol concentrations do not appear to vary with on SIC itself (Fig. 9). The presence or absence of sea ice at the time of sampling appears to influence methanol concentrations more strongly.] This is further supported by the fact that higher methanol seawater concentrations were measured in the relatively ice-free Smith Sound (46 nmol dm$^{-3}$) and Lancaster Sound (41 nmol dm$^{-3}$),*

compared to the more ice covered west Baffin Bay (23 nmol dm$^{-3}$). Our underway methanol measurements support dominant biological cycling of methanol, while oxidation rates probably exert a strong influence on dissolved concentrations.

*Sec. 4.2 Acetone*

*[Higher concentrations of acetone in partially sea ice covered ocean could be due to exposure of photolabile organic carbon from under the sea ice and the influence of sea ice on light penetration depth, thus further supporting a dominant photochemical source of acetone during this cruise track.] In support of this, we observe in the mean higher concentrations of acetone in Smith Sound (10.8 nmol dm$^{-3}$) and Lancaster sound (11.7 nmol dm$^{-3}$), which are both relatively open water areas downstream of sea ice edges, compared to West Baffin Bay (3.4 nmol dm$^{-3}$).*

*Sec. 4.3 Acetaldehyde*

*Similarly, we observe higher concentrations of acetaldehyde in Smith Sound (5.5 nmol dm$^{-3}$) and Lancaster Sound (6.0 nmol dm$^{-3}$), compared to West Baffin Bay (1.1 nmol dm$^{-3}$). Smith and Lancaster Sound are both areas where unbleached organic carbon is exposed to light as it is moved by ocean currents from ice covered waters into ice-free waters. This leads to higher production of acetaldehyde in these areas.*

*Sec. 4.4 Relationships between oxygenated VOCs*

*All three oxygenated VOCs (methanol, acetone and acetaldehyde) measured during this cruise generally display lower concentrations during the first week of sampling, which corresponds to sampling the sea ice zone of the more marine-influenced West Baffin Bay area. The slightly higher concentrations of these compounds nearer to land, i.e. in Smith Sound and Lancaster Sound, may be related to terrestrial sources or production of these gases as water masses are exposed to ice-free conditions by ocean currents. Methanol, acetone and acetaldehyde display gradually increasing concentrations as the vessel transects towards the ice edge in Lancaster Sound between 04/08 and 06/08.*

*Sec. 4.5 DMS*

*In the mean, higher concentrations of DMS were measured in West Baffin Bay (1.61 nmol dm$^{-3}$) and Smith Sound (1.59 nmol dm$^{-3}$) compared to Lancaster Sound (1.00 nmol dm$^{-3}$). It could be that these differences are due to different stages of the phytoplankton bloom at the different locations. At the same time, it is interesting that Smith Sound displays slightly higher concentrations of DMS compared to Lancaster Sound. This could be because the ice behind the ice bridge in Lancaster Sound tends to be multiyear ice (McLaughlin et al., 2004) which leads to different phytoplankton bloom and DMS dynamics, producing higher DMS concentrations further downstream, compared to first year ice edges (Abbatt et al., 2019; Lizotte et al., 2020).*

*Sec. 4.6 Isoprene*

*Average concentrations of isoprene were very similar in West Baffin Bay, Smith Sound and Lancaster Sound. It appears that other factors such as sst, Chl a and SIC at the time of sampling affect isoprene concentrations more strongly than different ocean dynamics in these areas.*

*Sec. 5 Air-sea fluxes*

*On average, we observe higher mean DMS fluxes in Smith Sound (1.44 µmol m$^{-2}$ d$^{-1}$), compared to West Baffin Bay (1.12 µmol m$^{-2}$ d$^{-1}$) and Lancaster Sound (0.52 µmol m$^{-2}$ d$^{-1}$). [...] Indeed, isoprene*

*fluxes were higher on average as well in Smith Sound (0.065 µmol m⁻² d⁻¹), compared to West Baffin Bay (0.041 µmol m⁻² d⁻¹) and Lancaster Sound (0.029 µmol m⁻² d⁻¹).*

**Specific comments**

P1, line 21 - …broadly higher concentrations… This is vague. Any numbers or statistics to support this interpretation?

*In the main text, mean concentrations with and without sea ice cover as well as linear correlations are presented. This sentence has been changed to:*

*Underway (3-4 m) measurements showed higher concentrations in partial sea ice cover compared to ice-free waters, for most compounds.*

P1, line 31 - …once the ice has melted… Or when under-ice water masses move out of the ice pack (see General Comment)

*The following sentence has been added to the abstract to summarise this:*

*Differences in underway concentrations based on sampling region suggest that water masses moving away from the ice edge influences dissolved gas concentrations.*

P11, line 194 - …Here we briefly discuss the effect of sea ice concentration (AND WATER CIRCULATION)… Water masses circulation at the ice edge is also important if one want to understand the impacts of ice edges on water biogeochemistry.

*We agree with the reviewer, though in this specific section, we focus on depth profiles. We only discuss the effect of water circulation on underway surface water measurements. See also our reply to the general comment and how this has been addressed.*

P11, line 202 - …could be characterised… Since not measured during this cruise.

*The manuscript has been changed here according to reviewer's suggestion.*

P11, line 207 - …for this region… Which region? West Baffin Bay? Lancaster Sound? The area covered during this study encompassed different 'regions', even if I admit the that term 'region' is vague…

*We replaced "for this region" by "for the Canadian Arctic Archipelago".*

P11, line 208 - …that that…(typo)

*Typo corrected.*

P11, line 213 - …in this region… Same comment as above. Please be more specific about the localisation.

*We replaced "in this region" by "in the Canadian Arctic Archipelago".*

P12, line 227 - …These may be in part due to under ice phytoplankton blooms… Under ice phytoplankton blooms take place 'under the ice', so this statement is confusing in respect of the previous sentence stating that Chl a is lower at full ice cover.

*This statement has been changed to:*

*These may be in part due to ice-edge blooms …*

P12, line 229 - …these compounds… Which ones? All the compounds measured during this study?

*This has been replaced by:*

*… the compounds measured in this study, …*

P 14, line 271 - …could be… The rest of the sentence is missing.

*Thank you for spotting this. The sentence has been completed to read:*

*Another reason for higher methanol concentrations during this cruise could be slower bacterial consumption which has been shown to vary seasonally (Sargeant et al., 2016).*

P14, line 275 - …phytoplankton SPECIES…¸

*The manuscript has been changed here according to reviewer's suggestion.*

P14, line 277 - …to be highly variable… I suggest deleting the (a), (b) and (c) since this style is not used elsewhere in the manuscript.

*We deleted the (a), (b) and (c).*

P14, line 281 - …concentrations more strongly… This paragraph will benefit to have a clear concluding sentence.

*A concluding sentence has been added:*

*Our underway methanol measurements support dominant biological cycling of methanol, while oxidation rates probably exert a strong influence on dissolved concentrations.*

P15, line 309 - …it appears that photochemistry… The present data set cannot identify processes (sources/sinks) at play since no rate measurements were conducted. This conclusion is not backed by observations.

*This sentence has been changed to:*

*While it is possible that some of the acetone we observed below ≈ 10 m is derived from biological activity, the near surface gradient of acetone concentration suggest that photochemistry is the dominant source of acetone in the upper 10 m.*

*Additionally, we introduced this possibility to the reader at the start of the discussion:*

*Additionally, we speculate in this section on the dominant processes (photochemistry or biologial source or sink) based on variations in concentrations. This speculation could have been more conclusive if we had made concurrent rate measurements.*

P16, line 323 - …SLIGHTLY higher concentrations of acetone… For clarity I suggest to had 'slightly' since the difference in the mean concentrations of acetone between ice and ice-free waters, although statistically significant, is small.

*The manuscript has been changed here according to reviewer's suggestion.*

P16, line 341 - …which also SHOW… Replace 'observe' by 'show'.

*The manuscript has been changed here according to reviewer's suggestion.*

P16, line 344 - …by freshwater input from melting sea ice…AND RIVERINE INPUT As mentioned in the Introduction (P 3, line 73), river runoff is also an important source of turbidity in the Arctic

*The manuscript has been changed here according to reviewer's suggestion.*

P17, line 376 - …These RESULTS suggest common sources…

*The manuscript has been changed here according to reviewer's suggestion.*

P17, line 376 - …qualities are poor… I suggest deleting 'quite'.

*The manuscript has been changed here according to reviewer's suggestion.*

P18, line386 - …The slightly higher…may be… This interpretation is very speculative and not supported by the data. In addition, the two sampled regions are separated by many kilometers and were not sampled at the same time. Other processes may be at play.

*Indeed, the discussion has been changed to reflect water circulation patterns and the presence of ice edges – see our response to the general comment.*

P18, line 390 - …This could be related to phytoplankton at the bottom of the ice… Replace 'phytoplankton' by 'ice algae'.

*The manuscript has been changed here according to reviewer's suggestion.*

P18, line 385 (line 397) - …We generally observed… This last statement at the end of the paragraph needs to be discussed. Why mentioning that here and what are the implications?

*A discussion has been added here:*

*We generally observe similar concentrations of DMS at 2 m and at 30 cm, except in near full ice cover (75 to 90 % SIC) (Fig. 5a, d) where concentrations at 30 cm are slightly higher than at 2 m, possibly due ice algae and the associated microbial web rapidly producing DMS.*

P18; line 400 - The references to Mungall et al. and Abbatt et al. are appropriate, but the authors should also refer to the more detailed and ocean-focused paper by Lizotte et al. (2020), Biogeosciences.

*A reference to Lizotte et al. (2020) has been added here.*

P18, line 402 – The authors should also compare their results with those reported in the recently published paper by Galí et al. (2021) for the western Baffin Bay ice edge zone. Galí et al. DMS emissions from the Arctic marginal ice zone. Elementa: Science of the Anthropocene (2021) 9 (1): 00113.

*Thank you for pointing us to this reference. We added a few references to this publication in this section.*

P18, line 408 – Refer to Lizotte et al. (2020).

*A reference to Lizotte et al. (2020) has been added here.*

P18, line 411 – Excluding...however. I suggest deleting this part of the sentence and to directly state: 'No significant correlation could be observed BETWEEN DMS CONCENTRATIONS AND SEA ICE CONCENTRATIONS DURING THIS STUDY.'

*The manuscript has been changed here according to reviewer's suggestion.*

P18, line 413 - …dependent on biological settings, presence/absence of under-ice bloom, water masses circulation in respect to the ice edge, time of the year, and type of sea ice.

*The manuscript has been changed here according to reviewer's suggestion.*

P19, line 423 - …This frequently… Any statistics?

*This has been changed to:*

*In the casts shown in Fig. 7, this frequently coincides with higher oxygen concentrations at the same depth, suggesting that gases produced at this depth from biological activity are not efficiently vented to the atmosphere.*

P19, line 432 - …This suggests…

*The manuscript has been changed here according to reviewer's suggestion.*

P19, line 442 - …have suggested… Replace 'suggested' by 'calculated'.

*The manuscript has been changed here according to reviewer's suggestion.*

P20, line 454 – …These depth profiles… What is mentioned in this paragraph is relevant to all gases measured in this study. So, it should not be at the end of the section on Isoprene. I see two options: the concerns mentioned here could be addressed in the different sections of the Discussion as relevant, or the whole paragraph (with some modifications) could be moved to the very beginning of the discussion as a warning.

*The whole paragraph has been moved to the beginning of the discussion with some modifications. It reads now like this:*

*The depth profiles and underway data discussed in the following section represent measurements at different times and locations. Therefore, differences are possibly not only due to sea ice coverage but could also be due to the oceanography of the area (McLaughlin et al., 2004). We recognise that sea ice is a very heteogenous environment with respect to ice thickness (Hayashida et al., 2020), the presence of melt ponds (Gourdal et al., 2018; Park et al., 2019), and types of sea ice (e.g. first year vs. multiyear ice (Lizotte et al., 2020)). This heterogeniety likely leads to very dfferent biogeochemistry, affecting trace gas cycling. Most of the discussion that follows here focusses largely the effect of on sea ice concentration on these gases and does not always explicitly take into consideration the effect of these variables, which is worthy of future research.*

*Additionally, we speculate in this section on the dominant processes (photochemistry or biologial source or sink) based on variations in concentrations. This speculation could have been more conclusive if we had made concurrent rate measurements.*

P23, line 535 - …impact the overlying atmosphere…BY… This idea should be further developed. What will be the impact?

*We decided to delete this sentence as, the claim made in this sentence is beyond the scope of this manuscript.*

END

---

## Author Comment (AC3)

**Reply to Reviewer 3 comments for: "Sea ice concentration impacts dissolved organic gases in the Canadian Arctic" by Charel Wohl et al.**

Many thanks to the reviewer for taking the time to thoroughly review the manuscript and provide constructive comments. The reviewer provided thought provoking comments, which has helped us to improve the manuscript. Please see our responses below. Reviewer comments are in normal font and author's replies can be found in italic.

Within the present publication a unique data set of arctic sea water measurement of methanol, acetaldehyde, acetone, DMS and isoprene is presented by the authors. These measurements were conducted during 17/07201 to 08/08/2017 onboard of CCGS Amundsen. The measurements are distinguished between different sea ice cover periods and thus provide a very interesting insight in how sea ice cover is able to influence production of organic materials relevant for the atmosphere. From the measured sea water values the corresponding emission fluxes are calculated afterwards. The results of the paper fit very well into the scope of Biogeosciences.
The paper is well structured and the results are logically discussed. A deep discussion of the results with measured values from literature is done, too. However, I think in some parts more discussion is mandatory, especially in the conclusion on the atmospheric oxidation capacity. Furthermore, I find the figures 2 to 6 hard to interpret. I recommend publication after the addressing of my questions and comments.

**General comments**

I find it hard to understand the figures 2 to 6 in which the depth profiles are presented. How am I able to know what concentrations was measured? In the legend are bars sketched that represent a certain concentration range, but these are not presented in the figure. These have to be added for the measurement points, otherwise the further discussion cannot be well comprehended.

*Many thanks to the reviewer for highlighting this. The casts have been grouped by sea ice concentration and horizontally offset to show the shape of the dissolved gas distributions with depth. A lot of the discussion focusses on the shape, rather than absolute concentrations. We recognize that it can be difficult to see what absolute concentration has been measured at each depth. We thus agree to add mean vertical profiles grouped according to ice cover and binned in different depth horizons. These mean vertical profiles show mean absolute concentrations at different depths for different sea ice concentrations.*

*The following introductory text has been added to the manuscript:*
*In Figures 2 to 6, panels (d) to (e) indicate mean vertical profiles grouped according to ice cover and binned in different depth horizons. These mean vertical profiles show mean absolute concentrations at different depths for a range of sea ice concentrations. Binning depth horizons were as follows; 0-0.5, 0.5-4, 4-10, 10-20, 20-30, 30-40, 40-50 and 50-60 m. Smaller bins were chosen near the surface to investigate the near surface gradients.*

*Figures 2 to 6 have been updated to include a mean vertical profile for different sea ice concentrations. Here we share the updated Figure 2 only to avoid over-crowding this document. References to the additional panels (d) to (f) have been added to the discussion text.*

[Figure]

**Figure 1 Overview plot displaying the shape of all methanol and density ($\sigma_T$) depth profiles, grouped by SIC and staggered along the x-axis for ease of viewing. Panel labels indicate the SIC bin. The scale bars for methanol and density in panel (a) apply also to panels (b) and (c). Profiles with hollow markers are highlighted in** Error! Reference source not found.**. Sampling dates are indicated to locate stations using Fig. 1. Panels (d) to (f) indicate absolute concentrations of mean vertical profiles grouped according to ice cover and binned by depth horizons. The shaded area indicates standard error for each depth horizon.**

Line 271
Here discussion text is missing.

*Thank you for pointing this out. Discussion text has been added in this place:*

*Another reason for higher methanol concentrations during this cruise could be slower bacterial consumption which has been shown to vary seasonally (Sargeant et al., 2016).*

It would be worth to compare the measured DMS values also with the values in Lana et al. (2011) and Hulswar et al. (2021). These are often used in global models to determine the effect of DMS on climate. I suggest a small discussion of these data in comparison with the measurements due to the possible benefit for the model community.

*A small discussion of our measured DMS values comparing them to these global databases has been added and is copied here:*

*An updated global climatology for DMS (Hulswar et al., 2021) predicts around 2 nmol dm$^{-3}$ for this sampling area during our sampling months, while the previous climatology (Lana et al., 2011) predicted around 2.5 nmol dm$^{-3}$. We note that the updated climatology includes new measurements in this sampling area, but still does not reflect these more recent and very high measurements of DMS in the sea ice zone cited above.*

Line 437 and following
In Dani and Loreto (2017) it was stated that "globally (i) marine phytoplankton taxa tend

to emit either DMS or isoprene, and (ii) sea-water surface concentration and emission hotspots of DMS and isoprene have opposite latitudinal gradients". The results presented here reveal that this might not be true for oceans in interaction with sea ice, and coastal areas. A small discussion has been already done, but I miss a bit one in regards to the statement of Dani and Loreto (2017).

*A small discussion has been added in regards to how our measurements fit with the statement by Dani and Loreto (2017). It has been copied over here:*

*Dani and Loreto (2017) suggest opposite latitudinal distributions of isoprene and DMS, with higher concentrations of DMS and lower concentrations of isoprene at the poles. Our data shows surprisingly high isoprene concentrations which does not fit this trend. It is possible that the statement by Dani and Loreto (2017) does not hold in the Arctic, where we suspect that terrestrial influence or the effect of sea ice leads to high isoprene concentrations.*

Regarding the emission calculation of DMS I think it is not so easy to neglect the gasphase concentration. For gas-phase DMS in the Northern Atlantic up to 35 ppt and in the Antarctic more than 200 ppt were measured (see review of Yu and Li, 2021). The Henry's Law coefficient for DMS is 1.55 at 273°K. This results into a steady state water concentration of 0.05 and 0.31 nmol dm-3, which is 4% and 22% of the mean measured sea water value. Therefore, it is my opinion that for DMS the gas-phase concentration cannot be easily neglected without some bias that has to be discussed.

*We agree with the reviewer and we decided to recompute our DMS flux. To recompute our DMS flux, we assume 185.5 pptv of atmospheric DMS as measured previously measured in the mean by Mungall et al. (2016) during a similar time of year and at a similar location.*

I think the discussion of the assessment of acetone and methanol towards the lifetime of methane and other pollutants has to be deeper. Regarding the applied background values of methanol and acetone in this study together with the 1.8 ppm methane the first order reaction rate of methane with the OH radical is more than 2000% higher than that of methanol and acetone combined. The possible higher emission rates of DMS and isoprene might have a stronger effect. This discussion has to be done more deeply.

*A discussion of the effect of these emissions is beyond the scope of this paper as indeed, we have made no measurements or modelling to comment on the effect of these emissions. We agree though that in some instances we have not made this very clear. We only suggest that our measurements will be useful to assess the impact of these fluxes on the oxidative capacity of the atmosphere. We have subsequently removed any suggestions that our methanol and acetone fluxes affect lifetimes of methane and other pollutants.*

**Minor comments**
Line 124
What method has been used to correct the 16m wind speed to the 10 m neutral wind speed?

*We converted the 16 m ASL Gill wind speed to 10 m ASL using the following standard method based on a logarithmic wind profile:*
$U_{10} = (u_*/\kappa) \ln(10/z_0)$

where $u_* = (\tau / \rho)^{1/2}$ is friction velocity (m s$^{-1}$), $\kappa$ is the von Karman constant of 0.4, z is measurement height, and $z_0$ is roughness length (m) calculated as
$z_0 = z \exp[-\kappa U(z)/u_*]$

This has been changed in the manuscript to:
Wind speed was measured from a meteorological tower (approximately 16 m) located on the foredeck of the ship, similar to that described in Ahmed et al. (2019). The measured wind speeds at 16 m above sea level were converted to 10 m wind speed (U10) based on a logarithmic wind profile (Kaimal and Finnigan, 1994) and corrected for speed of ship passage.

I would recommend to add a table into the supplement that displays the physico-chemical characteristics of the gases for the air-sea flux calculation.

A table has been added to the supplement listing mean physico-chemical characteristics required for the air-sea flux calculation. We have copied it over in this document as well:

**Supplement S4**

A table listing the cruise mean physico-chemical characteristics required for the air-sea exchange calculation is displayed here.

Table S1: Mean physico-chemical characteristics required for the air-sea exchange calculation. The values presented here have been calculated for the mean seawater temperature of 1.2 °C, and a cruise mean wind speed at 10 m (U10) of 4.8 m s$^{-1}$. Henry solubility values are defined here as dimensionless water over gas solubility in ambient seawater. Waterside Schmidt numbers have been calculated using the supplementary R code from Johnson (2010) at a seawater salinity of 35. Waterside transfer velocities ($k_w$) for isoprene have been calculated using the equation from Nightingale et al. (2000). Waterside transfer velocities for methanol, acetone and DMS have been calculated using the parametrisation by Yang et al. (2011). Airside transfer velocities for methanol, acetone, DMS and isoprene have been computed using the equation proposed by Yang et al. (2013).

|  | methanol | acetone | DMS | isoprene |
|---|---|---|---|---|
| $H$ /(1) | 14215.6 | 2010.2 | 28.5 | 1.4 |
| $S_{Cw}$/(1) | 2364.2 | 3403.2 | 3386.8 | 3723.7 |
| $k_w$/(cm h$^{-1}$) | 3.7 | 3.1 | 3.1 | 3.0 |
| $k_a$/(cm h$^{-1}$) | | 1501.84 | | |

Line 375
I suggest to delete are significant, as this has already been stated.

The manuscript has been changed here according to reviewer's suggestion.

References added by the reviewer

Lana, A., et al. (2011), An updated climatology of surface dimethylsulfide concentrations and emission fluxes in the global ocean, Global Biogeochem. Cycles, 25, GB1004,

doi:10.1029/2010GB003850.

Hulswar, S., et al. (2021), Third Revision of the Global Surface Seawater Dimethyl Sulfide Climatology (DMS-Rev3), Earth Syst. Sci. Data Discuss. [preprint], doi:10.5194/essd-2021-236, in review.

Dani and Loreto (2017), Trade-Off Between Dimethyl Sulfide and Isoprene Emissions from Marine Phytoplankton, Trends in Plant Science, 22, 361-372, doi: 10.1016/j.tplants.2017.01.006.

Yu and Li (2021), Marine volatile organic compounds and their impacts on marine aerosol – A review, Sci. Total Environ., 768, 145054, doi: 10.1016/j.scitotenv.2021.145054.

*Additional References added as reply to reviewer's comments*

Mungall, E. L., Croft, B., Lizotte, M., Thomas, J. L., Murphy, J. G., Levasseur, M., Martin, R. V., Wentzell, J. J. B., Liggio, J. and Abbatt, J. P. D.: Dimethyl sulfide in the summertime Arctic atmosphere: Measurements and source sensitivity simulations, Atmos. Chem. Phys., 16(11), 6665–6680, doi:10.5194/acp-16-6665-2016, 2016.